# From Synapses to Dynamics: Obtaining Function from Structure in a Connectome Constrained Model of the Head Direction Circuit

**Sunny Duan**[*]
Brain and Cognitive Sciences
Massachusetts Institute of Technology
`sunnyd@mit.edu`

**Ling Liang Dong**[*]
Brain and Cognitive Sciences
Massachusetts Institute of Technology
`loading@mit.edu`

**Ila Fiete**
Brain and Cognitive Sciences
Massachusetts Institute of Technology
`fiete@mit.edu`

## Abstract

How precisely does circuit wiring specify function? This fundamental question is particularly relevant for modern neuroscience, as large-scale electron microscopy now enables the reconstruction of neural circuits at single-synapse resolution across many organisms. To interpret circuit function from such datasets, we must understand the extent to which the measured structure constrains dynamics. We investigate this question in the *Drosophila* head direction (HD) circuit, which maintains an internal heading estimate through attractor dynamics that integrate self-motion velocity cues. This circuit serves as a sensitive assay for functional specification: continuous attractor networks are theoretically known to require finely tuned wiring symmetries, whereas connectomes omit key cellular parameters such as synaptic gains, neuronal thresholds, and time constants, and reveal that biological wiring can be heterogeneous. We introduce a method that combines self-supervised and unsupervised learning objectives to estimate unknown parameters at the level of cell types, rather than individual neurons and synapses. Starting from the raw connectivity matrix, our approach recovers a network that exhibits continuous attractor dynamics and accurately integrates a range of velocity inputs, despite minimal parameter tuning on a connectome that notably departs from the symmetric regularity of an idealized ring attractor. We characterize how deviations from the original connectome shape the space of viable solutions. We also perform in-silico ablation experiments to probe the distinct functional roles of specific cell types in the circuit, demonstrating how connectome-derived structure, when augmented with minimal, biologically grounded tuning, can replicate known physiology and elucidate circuit function.

## 1 Introduction

Recent advances in large-scale electron microscopy have enabled the reconstruction of neural circuits at synapse-level resolution, producing dense connectomic datasets across organisms including *C. elegans* (Cook et al., 2019), *Drosophila* (Scheffer et al., 2020b; Dorkenwald et al., 2024), and mice (MICrONS Consortium, 2025). These datasets raise several fundamental questions: How precisely

---

[*]Equal contribution

39th Conference on Neural Information Processing Systems (NeurIPS 2025).

can circuit function be determined from its wiring? Can we do so in the absence of key membrane and synaptic parameters such as cellular thresholds, gains, and time constants? Additionally, connectomes provide detailed synaptic connectivity maps but may contain potential errors from the data pipeline, including misalignment of electron microscopy sections and false positives or negatives in synapse detection Scheffer et al. (2020a). Thus, individual connectomes represent partial and sometimes noisy snapshots of the underlying biological networks that may fail to capture the necessary factors that play a role in shaping circuit behavior.

We explore these questions in the *Drosophila* head direction (HD) circuit, a canonical example of a continuous ring attractor network in biology (Kim et al., 2017; Green et al., 2017; Turner-Evans et al., 2017) that maintains an internal estimate of animal heading by integrating angular self-motion velocity inputs. Whereas theoretical studies of ring attractors (Skaggs et al., 1994; Zhang, 1996; Redish et al., 1996; Xie et al., 2002) show that they require finely tuned connectivity to support bump stability and smooth translation for velocity integration, we found that connectomic reconstructions of this circuit exhibit asymmetries and heterogeneities. Moreover, the *Drosophila* HD circuit involves $\sim 10$ distinct cell types — many more than are theorized to be necessary in theoretical ring attractor models — raising open questions about the necessity and roles of specific circuit components.

Previous models of the HD circuit (Chang et al., 2023; Stentiford et al., 2024) have relied on hand-designed circuit architectures that impose connectivity motifs, in particular circular symmetry, known from earlier work to support ring attractor dynamics (Skaggs et al., 1994; Zhang, 1996; Redish et al., 1996; Xie et al., 2002). While such models produce the expected network dynamics by design, they start with prior knowledge about circuit structure and the identities of the cell types involved in the core circuit, thus limiting opportunities for data-driven discovery. Here, we introduce a framework that bridges this gap: starting from the original measured connectivity matrix, we optimize only a small set of biologically grounded parameters at the cell-type level, such as cell-type-to-cell-type synaptic gains and cell type-shared neuronal biases and time constants. The model is trained using a self-supervised linear consistency loss that simply enforces the internal bump movement to be proportional to the input velocity, in addition to several regularization terms that ensure the model learns a non-trivial solution that remains close to the original connectome. Notably, this learning objective requires no neural activity recordings or behavioral labels, enabling inference of functional dynamics from only structural measurements.

Our trained model recovers high-fidelity continuous attractor dynamics and robustly performs angular velocity integration across a wide range of inputs. It also reproduces experimental findings including realistic bump width and bidirectional bump motion dependent on asymmetric input from the left and right P-EN subpopulations (Seelig and Jayaraman, 2015; Turner-Evans et al., 2017). We find that tuning parameters at the cell-type level achieves performance comparable to models with more parameters, while global parameter tying fails to recover attractor behavior—indicating that cell-type–specific tuning offers a practical tradeoff between model flexibility and parsimony, performing comparably to fully parameterized models while preserving biological interpretability. We also explore the space of viable solutions under different levels of simulated noise and characterize the diversity of model solutions as a function of their initial starting points on the loss landscape. Finally, we conduct in silico ablation experiments to probe the roles of different cell types.

**Key Contributions**

- We introduce a self-supervised connectome-constrained framework that recovers functional dynamics of integrator circuits directly from noisy connectome measurements

- We show that parameterization at the level of cell types is sufficient and necessary to recover integrating network dynamics

- We characterize how initial conditions shape the diversity of learned solutions.

- We perform in silico ablations on the trained networks, yielding novel biological insights on the functional roles of specific neuron classes in the Drosophila HD circuit.

## 2 Background

### 2.1 Continuous Attractor Models

Continuous attractor networks encode analog variables, such as orientation or position, via localized bumps of persistent activity that shift smoothly across the neural population. These dynamics require precise tuning of recurrent connectivity, typically balancing local excitation and broad inhibition (Khona and Fiete, 2022). Even minor deviations in synaptic strength or timing can cause bump instability, drift, or collapse. This sensitivity makes such networks a strong testbed for evaluating the extent to which connectomes alone can specify circuit dynamics.

### 2.2 Drosophila HD Circuit

The Drosophila HD circuit resides in the central complex and comprises several cell types (populations of cells that exhibit shared morphological, genetic, and connectivity profiles) thought to implement a ring attractor (Kim et al., 2017; Turner-Evans et al., 2020). E-PG neurons in the ellipsoid body (EB) are connected in an anatomical ring-like structure and comprise the core set of cells that track animal heading via a localized bump in population activity (Seelig and Jayaraman, 2015). P-EN neurons convey angular velocity signals: left and right P-EN neurons asymmetrically project to E-PG neurons to shift the bump based on the directional velocity, and are further subdivided into P-ENa and P-ENb subpopulations whose differential roles in the circuit have yet to be ascertained (Turner-Evans et al., 2017). Inhibitory inputs from Delta7 and ring neurons are believed to normalize and gate activity (Green et al., 2017; Hulse et al., 2023), while P-EG neurons are thought to contribute to bump stabilization (Pisokas et al., 2020), although this has not yet been confirmed experimentally. These cell types outnumber those in idealized ring attractor models (Skaggs et al., 1994; Zhang, 1996; Redish et al., 1996; Xie et al., 2002), which typically define a core set of two to three cell types to implement local excitation, global inhibition, and bump movement, raising open questions about the role and necessity of each cell type.

For our experiments, we use connectomic data from the Drosophila FlyEM hemibrain dataset (Scheffer et al., 2020b) after a simple preprocessing step that takes advantage of the natural symmetry between the left and right brain hemispheres to correct synapse counts through a symmetrization process (Section A.1). We focused our analysis on a subpopulation of neurons previously implicated in head direction (HD) processing, comprised of a population of 439 neurons spanning six cell types: E-PG, Delta7, P-EG, P-EN, GLNO, and ring neurons. Notably, the ring neuron population, which is responsible for conveying sensory inputs to the circuit, are divided by lineage into ER and ExR subtypes, which can further be divided into a diverse set of 29 different subtypes of ring neurons.

The connectome provides signed synapse counts (where signs are inferred based on measured neurotransmitters (Davis et al., 2020)) but lack key parameters such as synaptic gains or time constants, and include asymmetries that likely result from biological and measurement noise. This motivates the need for methods that can infer minimal functional parameters from structure while remaining faithful to the measured connectivity.

## 3 Related Work

Connectome-based models have been used to study neural circuit function across a range of scales and assumptions. Chang et al. (2023) tested whether the required global inhibition in the fly head-direction (HD) circuit is provided by Delta7 or ring neurons, using spiking models hand-constructed from connectome-derived motifs and evaluating bump-based metrics across manual parameter sweeps. Stentiford et al. (2024) built a detailed spiking model of the central complex using prespecified physiological parameters, connectivity, and receptive fields, and showed that the models could learn visual cue-to-heading associations via Hebbian plasticity. Pospisil et al. (2024) inferred a whole-brain "effectome" from optogenetic perturbation data, using the connectome as a structural prior along with full-brain activity recordings to infer the effective weights of a linear model. Other works (Mi et al., 2023; Lu et al., 2025) have leveraged neural recordings to infer connectivity structure. Our work is most similar to Lappalainen et al. (2024), who train a deep network constrained by the fly visual motion connectome. However, they assume an idealized, perfectly tiled columnar architecture, and optimize cell-type parameters via supervised learning on an optic flow task. In contrast, we employ

unsupervised training directly on the raw, measured connectome. Finally, though not a connectome-constrained model, Schaeffer et al. (2023) also used self-supervised objectives to train a circuit to learn an internal representation of a spatial variable, and introduce a conformal isometry loss similar in concept to our linear consistency loss to enforce proportional updates of neural representations with input magnitude. However, whereas they train a randomly initialized network with activity-level constraints, we instead initialize a network with measured connectomic structure, and train it to learn biologically consistent representations through cell-type-level parameterization and self-supervised loss terms without explicit neural activity constraints.

Our approach differs in several key ways. We operate directly on the raw connectome without architectural simplification, activity recordings, or external supervision. We introduce a self-supervised learning objective based on internal representational consistency, which enforces that changes in internal velocity reflect changes in input velocity without requiring activity measurements or task labels, and optimize only a small set of biologically interpretable, cell-type–level parameters. Our method recovers continuous attractor dynamics despite known physiological asymmetries and irregularities of the connectome. Because the objective encodes generic dynamical constraints rather than task-specific labels, our framework can be extended to other circuits hypothesized to implement a coarse computational function, such as integration, memory, or normalization, offering a scalable approach to functional inference from connectomic structure without activity recordings.

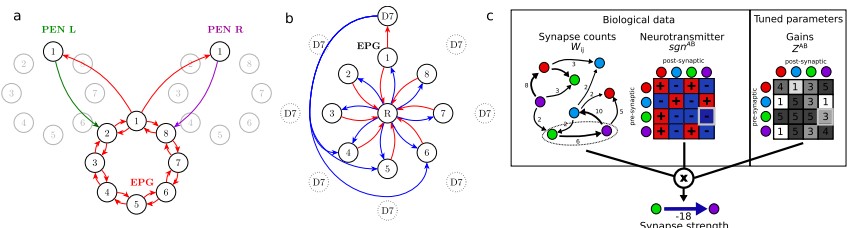

Figure 1: **The HD circuit and model setup. a.** The connectivity pattern of the PEN and EPG neurons. The left/right PEN neurons asymmetrically project to the EPG neurons, causing the ring to rotate clockwise or counterclockwise depending on net input drive. **b.** Ring neurons provide uniform inhibition to EPG neurons, while Delta7 neurons target EPG neurons far away on the ring. **c.** Schematic of tuning procedure. Fixed synapse counts and signs inferred from neurotransmitters are combined with trained cell-type-specific gain parameters to produce the final network weights. Nodes are colored by cell type.

## 4 Methods

### 4.1 Model description

**Rate-based neuron dynamics model** We modeled the dynamics of this neural circuit using a rate-based dynamics model. The firing rate $x_j$ of each neuron $j$ in the circuit evolves according to the following differential equation:

$$\tau \frac{dx_j(t)}{dt} = -\ell\, x_j(t) + \sigma\left(\sum_k W_{jk} x_k(t) + b_j + u_j(t)\right) \tag{1}$$

where $x_j(t)$ is the time-varying firing rate of neuron $j$, $\tau_j$ represents a global intrinsic time constant of each neuron and $\ell$ is a global constant determining the scale of activity and, together with $\tau$, the rate of activity decay in the absence of input. $\sigma(\cdot)$ denotes the sigmoid function, $W_{jk}$ represents the synaptic weight from neuron $k$ to neuron $j$. $b_j$ is the threshold parameter for neuron $j$, and $u_j(t)$ represents any (potentially time-varying) external inputs injected into neuron $j$.

**Connectomics-based model parameterization** To model conectomically defined circuits, we define the weights in Eq. 1 as $W_{ij} = w_0(1 + Z_{ij}) sgn_{ij} C_{ij}$ where the connectivity matrix ($C$) and the signs ($sgn \in \{\pm 1\}$) of the interaction are fully determined by the empirical synapse counts measured in the connectome and from recorded neurotransmitters, respectively (Scheffer et al., 2020b). The

value $w_0$ is a common gain factor that relates synapse count to connection strength; $Z_{ij}$ corresponds to an additional, unknown differential synapse-specific gain; changes in this parameter permit deviations from connectomically-defined structure. The connectome does not specify neural time constants, neural thresholds, and the neural non-linearity (transfer function). Thus, the parameters $\{Z, b, \tau\}$ are the subjects of optimization.

To examine the extent to which the connectome alone provides sufficient information for detailed dynamics, we constrained the optimization problem with biological inductive biases by assuming that the gain and bias parameters $Z, b$ are shared across all neurons of a given cell type. This parameterization at the level of cell types is biologically motivated, as neurons of the same type exhibit similar gene expression, intrinsic electrophysiological properties, and connectivity patterns (Davie et al., 2018; Turner-Evans et al., 2020). Thus, instead of optimizing parameters $Z_{jk}, b_j$ for each for each synapse and neuron, we optimized a much smaller set of cell-type-specific parameters $Z^{AB}, b^A$, where $A, B, \ldots \in \mathcal{C}$ index cell type. In other words, $Z^{AB}$ is the shared gain factor for the all synapses from neurons of Type B to those of Type A and $b^A$ is the threshold of all neurons of type $A$. The dynamics are therefore given by:

$$\tau \frac{dx_j^A(t)}{dt} = -\ell \, x_j^A(t) + \sigma \left( \sum_{B \in \mathcal{C}} \sum_k w_0 (1 + Z^{AB}) sgn^B C_{jk} x_k^B(t) + b^A + u_j(t) \right) \quad (2)$$

This reduces the number of optimized parameters from $439^2 + 439 + 1 = 193,161$ to just $7^2 + 7 + 1 = 57$ parameters, relative to a full per-synapse parameterization.

## 4.2 Task-based optimization

**Velocity integration task**   We optimize the free network parameters over a set of simulation trials using a set of biologically-motivated self-supervised and unsupervised objectives under the assumption that the circuit performs integration of its input signals.

**State initialization** For each simulation trial, we initialized the firing rates of a single E-PG neuron to 1, representing a bump on the ring, and initialized the activities of all other neurons to 0.

Dynamics were simulated according to Eq. 2 for a fixed duration of $T_{sim} = 2s$ using the Dormand-Prince method (Dopri5) (Dormand and Prince, 1980) provided in the Diffrax (Kidger, 2021) library in Jax. During each trial, a constant velocity input $u \in [-U, U]$ representing clockwise (negative), counter-clockwise (positive), or zero velocity) was applied to the network. Following experimental evidence that GLNO neurons provide angular velocity signals to the HD system (Hulse et al., 2023), the velocity input was selectively injected into either the left or right GLNO neurons.

**Training Objectives**   We define a set of biologically motivated objectives based on the assumption that the network is an integrator, i.e. that it should update its internal state as a linear function of the velocity input, and maintain its existing state in the absence of inputs. Notably, we do not specify particular activity profiles in the network, nor do we ask the state to change by a prespecified amount in response to specific velocity inputs.

In each trial, we evaluate the network state at discrete time points $t = (1, \ldots, T)\Delta t$ separated by interval $\Delta t$. We use population coding (Bialek et al., 1989) to decode the network state (location along the ring, $\theta(t)$) at each $t$ from the EPG neuron population by calculating the heading as the vector sum of neurons' preferred directions weighted by their firing rates. The change in the represented angle over the interval $\Delta t$ was computed as $\delta\theta(t) = \frac{\theta(t) - \theta(t-1)}{\Delta t}$.

- **Entropy**: Encourages diverse activation across the population by penalizing overly concentrated or uniform activity patterns.

$$\mathcal{L}_{\text{entropy}} = -\frac{1}{T} \sum_t^T \sum_j^N \phi_j(t) \log \phi_j(t)$$

  Where $\phi_j(t) = \frac{e^{x_j(t)}}{\sum_k^N e^{x_k(t)}}$ is the softmax of the neural activities at a given time point.

- **Stability**: Encourages stability, i.e., no changes in state given zero input ($u = 0$).

$$\mathcal{L}_{\text{stability}} = \frac{1}{T} \sum_t^T \delta\theta(t)^2 \cdot \mathbf{1}(u = 0)$$

- **Minimum Speed**: Prevents the model from learning a trivial zero solution.

$$\mathcal{L}_{\text{speed}} = -\frac{1}{T}\frac{1}{2U}\sum_{t}^{T}\sum_{u=-U}^{U}\delta\theta(t)^2\Theta(|u|)$$

where $\Theta$ is the heavy-side step function.

- **Linear Consistency**: Encourages the model to update its internal state as a linear function of the velocity input

$$\mathcal{L}_{\text{linearity}} = \frac{1}{T}\frac{1}{2U}\sum_{t}^{T}\sum_{u=-U}^{U}\left(\frac{\delta\theta(t)}{|u|} - \mu\right)^2\Theta(|u|)$$

where $\mu = \frac{1}{T}\frac{1}{2U}\sum_{t}^{T}\sum_{u=-U}^{U}\frac{\delta\theta(t)}{|u|}$

- **L1 and L2 Regularization**: Imposes costs for changes in weights changes, with the penalty size corresponding to the number of synapses that are impacted by each weight.

$$\mathcal{L}_{\text{L1}} = \sum_{i,j\,:\,i\in A,\,j\in B}|Z^{AB}|C_{ij}$$

$$\mathcal{L}_{\text{L2}} = \sum_{i,j\,:\,i\in A,\,j\in B}\left(Z^{AB}C_{ij}\right)^2$$

- **Total loss:**

$$\mathcal{L}_{\text{total}} = \beta_1\mathcal{L}_{\text{linearity}} + \beta_2\mathcal{L}_{\text{speed}} + \beta_3\mathcal{L}_{\text{entropy}} + \beta_4\mathcal{L}_{\text{L1}} + \beta_5\mathcal{L}_{\text{L2}} \tag{3}$$

Where $\beta_1, \beta_2, \beta_3, \beta_4, \beta_5$ are hyperparameters.

**Optimization**  The initial values of $Z^{AB}$ are all set to 0. Initial values of $b^A$ are set to a constant $b$ for all cell types. The quantities $w_0, b, \ell$ were chosen by hyperparameter optimization. All trainable parameters were optimized to minimize the total loss function $\mathcal{L}_{\text{total}}$ using Adam (Kingma, 2014).

## 5   Results

### 5.1   Robust integrator dynamics from minimal training of a connectome-constrained model

We optimize model parameters according to the procedure outlined above (Figure 2a). Allowing only a single gain factor ($w_0$) does not yield activity bumps; allowing two global gain factors (one for all excitatory synapses and another for all inhibitory ones) yields an activity bump but no integration. By tuning the small set of cell-type parameters defined above, we recover a dynamical system that is able to integrate bidirectional velocity inputs as well as maintain a stable heading representation in the presence of noise, displaying near-zero drift under constantly injected multiplicative noise at or below 1% of neuron activity (Figure 2b, c, d). Despite not specifying for output shape in our training, we find that the model produces an activity bump with a width (Figure 2e) close to previous experimental reports of $\pi/2$ (Kim et al., 2017). The model also accurately integrates velocity input, maintaining internal state updates that scale proportionally with input magnitude, and minimal pinning region (no bump movement to non-zero velocity inputs), enabling accurate integration over small velocity inputs and velocity direction changes (Figure 2f, g).

### 5.2   Tuned network exhibits characteristics of a ring attractor network

To evaluate whether the trained model exhibits dynamics characteristic of a continuous ring attractor, we analyzed the geometry of its internal state space. Principal component analysis of EPG activity during continuous, time-varying velocity inputs in both directions reveals that the network's population dynamics are constrained to a one-dimensional manifold (Figure 3a, b). This observation is confirmed by local intrinsic dimensionality analysis using local PCA (Kambhatla and Leen, 1997), which estimates the dimensionality of the dynamics to be approximately 1 (Figure 3c). The correlation dimension Grassberger and Procaccia (1983), which quantifies the global scaling properties of the attractor, is also close to 1 (Figure 3d), consistent with a smooth, low-dimensional integrator.

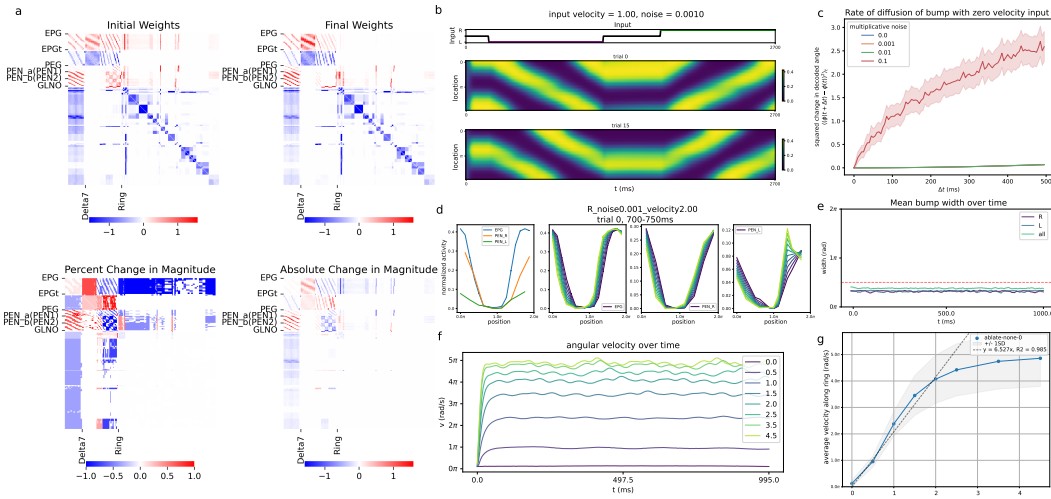

Figure 2: **Tuned network weights produce robust integrator dynamics. a.** Initial and trained network weights. **b.** The activity bump of the EPG neurons across no input and directional input conditions. **c.** The bump exhibits minimal drift for noise levels below 10% of neuron activity **d.** Slices of population activity for EPG and left/right PEN neurons, colored by time. Asymmetric activation of PEN populations drives bump movement in the corresponding direction. **e.** The width of the activity bump, when calculated over left, right, and all neurons, closely matches experimental measurements of $\pi/2$. **f.** The internal heading velocity increases proportionally to the magnitude of input velocity, indicating accurate integration. **g.** The input-output velocity relationship remains linear across a biologically plausible input range.

To further assess the topological structure of the dynamics, we apply persistent homology to the E-PG state trajectories using the Ripser package (Bauer, 2021; Tralie et al., 2018). This analysis yields a persistence diagram consistent with a 1D topological ring (Figure 3e), matching theoretical expectations for a circular continuous attractor. Together, these analyses demonstrate that the network does not merely perform velocity integration in the output space, but that its internal dynamics form a structured, low-dimensional ring manifold, in line with theoretical models of the *Drosophila* HD circuit.

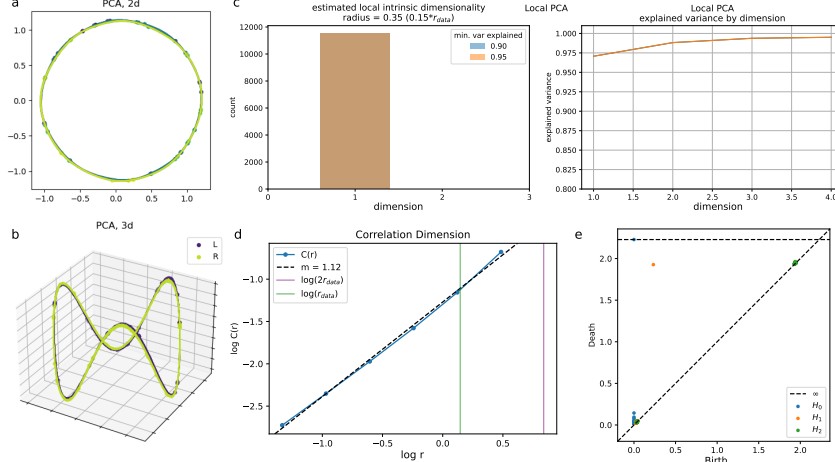

Figure 3: **State space geometry of the HD circuit a.** 2D projections of EPG activity via PCA show that the dynamics lie on a one-dimensional ring manifold. **b.** Ring manifold structure is preserved in a 3D projection in PCA space. **c.** Local PCA estimates the intrinsic dimensionality of the dynamics to be 1. **d.** The local correlation dimension of the network is also approximately 1. **e.** Persistent homology analysis yields a persistence diagram consistent with a 1D topological ring.

## 5.3 The connectome as a structural prior for functional dynamics

Given that finetuning the HD network by training cell-type parameters is sufficient to obtain a network that produces ring attractor dynamics despite noise in the connectome, we hypothesized that the connectome provides a structural prior that supports robust integrator dynamics. To test this, we trained models initialized with increasingly perturbed versions of the connectome. We found that small amounts of Gaussian noise, when preserving connection signs, still yielded integrating solutions. However, introducing sign flips degraded performance, and introducing more drastic perturbations, including shuffling weights within cell-type blocks as well as across the entire connectome, consistently failed to produce functional networks even after extensive training (Figure 4a).

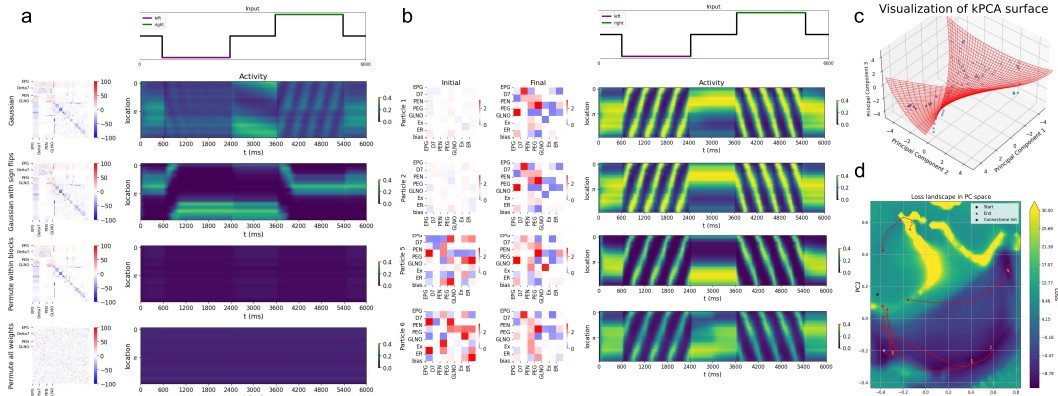

Figure 4: **Visualizing the space of solutions a.** From top to bottom: initialization weights and activity bumps of networks initialized with connectomes perturbed with gaussian noise ($\sigma^2 = 10$), gaussian noise with random sign flips ($p = 0.05$), permutation of weights within cell-type blocks, and permutation of all weights. **b.** Training runs with multiple initial weight matrix conditions. Left: Initial and learned cell-type weights ($Z^{AB}$) centered at $1.0$. Right: Heatmap depicting the neural activity over time when left and right inputs are given. **c.** Visualization of the surface fit by kernel PCA. The training trajectory points are used to fit the surface and are depicted by the blue dots. **d.** Visualization of training trajectories along the loss landscape. Trajectories are labeled similarly to b for comparison.

**Cell-type parameterization as the correct level of abstraction**  Although the connectome provides essential structural information, learning cell-type-specific parameters is critical for producing a functional network. We trained networks using simplified parameterizations by either tying parameters globally across all cell types, or by learning only broad excitatory and inhibitory cell type parameters (see Section A.2, Figure A4). Neither approach yielded a network capable of velocity integration. Overall, our results showed that a network capable of angular velocity integration does not emerge generically from any connectome, nor from arbitrary choices of biophysical parameters.

**Diversity of solutions from initial positions on the loss landscape**  We next explored the degeneracy of the loss landscape by adding different amounts of noise to the network parameters at initialization, and tracking their trajectories across training (Figure 4b). We visualize these trajectories on the fitted loss landscape (Section A.3) of solutions using kernel PCA (Schölkopf et al., 1998). We find that the final solutions of the network depend on the proximity of the initial parameters; networks close by tend to converge to solutions within the same basin, while networks initialized far apart tend to arrive at different solutions. Despite this divergence, networks that arrived at different solutions still produced functional integrator dynamics and reside in the same overall loss basin, indicating some degeneracy in the solution space but a common structural motif (Figure 4c, d).

## 5.4 Cell-type ablations

Given a functional model of the *Drosophila* head direction circuit, we next asked: what are the unique roles of each of the cell types within the circuit? To address this question, we performed ablation

experiments by zeroing the output weights of specific cell-type populations in our model. Because the model weights were optimized on the full set of cell types, we retrained the ablated networks to give each model a fair chance at recovering performance by altering the weights of the remaining cell types. Thus, we test which sets of neurons are strictly necessary in the sense that their function cannot be substituted by simply tuning the weights of other cell types within the network. As with the optimization of the original model, we sweep over a range of hyperparameters and select a set of the best performing models based on linear integration consistency and maximum bump velocity for evaluation (Section A.4). Across all experiments, ablated models exhibit less stability of the bump in the presence of large amounts of noise compared to the original network even after training (Figure 5).

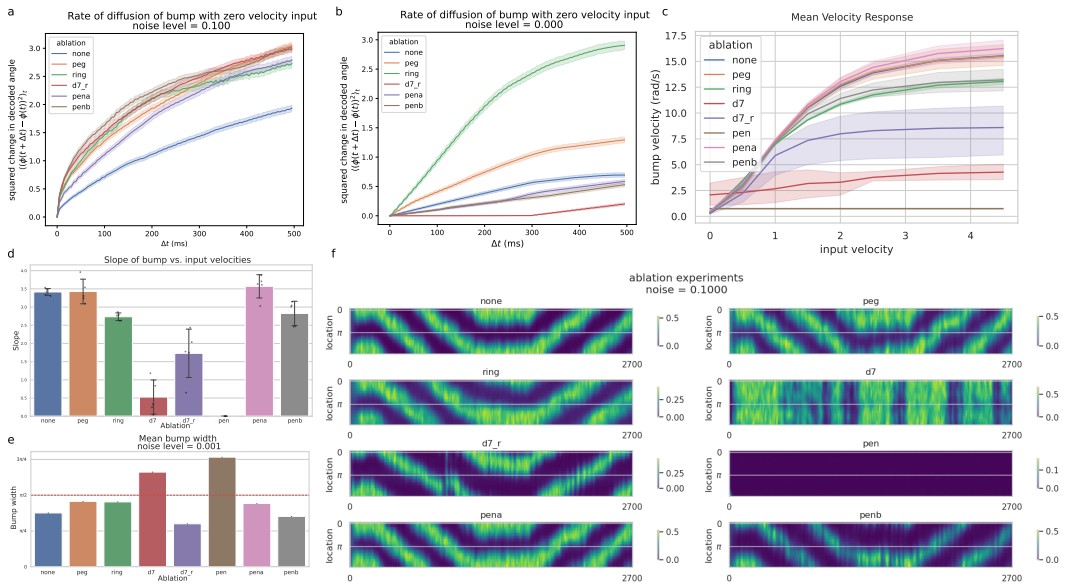

Figure 5: **Cell-type ablation experiments**. **a.** Rate of diffusion of the bump with zero inputs in the presence of multiplicative noise at 10% of neuron activity. Non-integrating solutions (Delta7 and P-EN ablations) not shown. **b.** Rate of diffusion in a noiseless condition. **c.** Mean bump velocity as a function of input velocity, colored by ablation. **d.** Slope of mean velocity response, calculated over inputs in the range of [0, 2]. **e.** Mean bump width of different ablation conditions. Red line indicates experimentally observed width. **f.** Sample EPG activity profiles by ablation, given the same input profile as in Figure 2.

Both Delta7 and ring neurons have been hypothesized to provide the crucial global inhibition to implement a ring attractor in the *Drosophila* (Chang et al., 2023). Although we found that ablating the ring neuron population reduced the range of represented velocities along the ring and displayed more drift in the presence of noise, ablating the Delta7 neurons had a far stronger effect: the network was no longer able to form a bump, let alone integrate velocity (Figure 5f). Notably, we found that the Delta7-ablated network was able to recover bump formation and integration when we relaxed the block matrix assumption and individually tune the weights of all the ring neuron subtypes (Section A.5), indicating that the *Drosophila* HD circuit relies on multiple types of inhibition, in contrast to theoretical ring attractor models.

The P-EG neurons have previously been hypothesized to play a role in stabilizing the bump in the *Drosophila* HD circuit (Pisokas et al., 2020), although experimental evidence in support of this is lacking. We find support for this hypothesis, as the P-EG-ablated model produces a network that exhibits more drift under even low amounts of noise even after training (Figure 5a, b).

Finally, although the role of the P-EN neurons in providing asymmetric drive to move the bump in the E-PG neuron population has been well characterized, the roles of the P-ENa and P-ENb subpopulations remain poorly understood. We confirm the role of the P-EN neurons by showing that a PEN-ablated network is unable to maintain or move a bump even after training (Figure 5f). We then separately ablated the P-ENa and P-ENb neuron subpopulations and found that while ablating

either subpopulation increased drift under noise, the P-ENb ablation had a stronger effect, leading to greater drift and a reduced range of bump velocities.

# 6 Conclusion and Discussion

We present a method that leverages biological priors and parameterization at the level of cell types to recover circuit dynamics from noisy connectome measurements using self-supervised learning. Without access to biophysical parameters, external task labels, or activity data, our model is able to recover robust continuous attractor dynamics capable of stable velocity integration in spite of observed asymmetries and small network size in the *Drosophila* head direction circuit. Interestingly, this result implies that connectomes, without biophysical gain parameters but with knowledge of connectivity signs, may be close to being lottery tickets (Frankle and Carbin, 2018). We characterize the solution space and show that multiple viable solutions emerge across different initializations, allowing estimation of parameter uncertainty through noise injection. We performed in silico experiments via targeted cell-type-specific ablations to elucidate the mechanistic roles of different cell types within the circuit.

Despite being able to recover circuit dynamics, the fidelity of the inferred parameters in this study to the underlying biological parameters may be limited in part by the fidelity of the connectome, because we performed additional (albeit minor) pre-processing of the connectivity matrix and we found a multiplicity of possible solutions. In addition, we made assumptions about neurotransmitter identity and synaptic signs based on cell types.

We believe our self-supervised learning approach will be powerful for future work on inferring function and activity from connectomes, relative to supervised approaches which require prior knowledge about circuit activity states. However, our stability and linear consistency losses terms assumed equivariance to inputs and involved population decoding, limiting their direct application to circuits that integrate continuous variables, such as oculomotor integrators (Seung, 1996), grid cells (Burak and Fiete, 2009), time cells (Kraus et al., 2013), or circuits involved in evidence accumulation (Shadlen and Newsome, 2001). However, we believe our framework could be easily extended by modifying the combination of losses used during training. An important future direction is to generalize our approach by exploring alternative unsupervised objectives that relax these assumptions. Additional future directions include extending the model to study multi-modal sensory integration with the internal state, and validating model predictions through closed-loop perturbation experiments.

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

# A    Technical Appendices and Supplementary Material

## A.1    Weight Symmetrization

To enforce bilateral symmetry in the neural network architecture while preserving functional connectivity patterns, we implemented a weight matrix symmetrization procedure. Let $\mathbf{W} \in \mathbb{R}^{N \times N}$ be the initial weight matrix, where $N$ is the number of neurons. For each neuron $i$, we define its lateralization $l_i \in \{L, R\}$ and its lateralized label $\tilde{l}_i$ (removing lateralization information).

The symmetrization process operates on connection groups defined by the tuple $(s, t, l_s, l_t)$, where $s$ and $t$ are the lateralized labels of the source and target neurons, and $l_s$ and $l_t$ are their lateralizations. For each unique connection group, we compute the symmetrized weight:

$$w_{sym}(s, t, l_s, l_t) = \frac{1}{2n_{st}} \sum_{i,j} w_{ij} \mathbb{I}[(s, t, l_s, l_t) = (\tilde{l}_i, \tilde{l}_j, l_i, l_j)]$$

where $n_{st}$ is the number of connections in the group and $\mathbb{I}$ is the indicator function. The final symmetrized weight matrix $\mathbf{W}_{sym}$ is constructed as:

$$W_{sym,ij} = \begin{cases} w_{sym}(\tilde{l}_i, \tilde{l}_j, l_i, l_j) & \text{if } \exists w_{sym}(\tilde{l}_i, \tilde{l}_j, l_i, l_j) \\ W_{ij} & \text{otherwise} \\ 0 & \text{if } i = j \end{cases}$$

This procedure ensures that left-right symmetric connections have identical weights ($W_{sym,ij} = W_{sym,ji}$ for lateralized pairs), while preserving the network's functional connectivity patterns and maintaining the overall network structure and strength. The result of this symmetrization is shown in Figure A1.

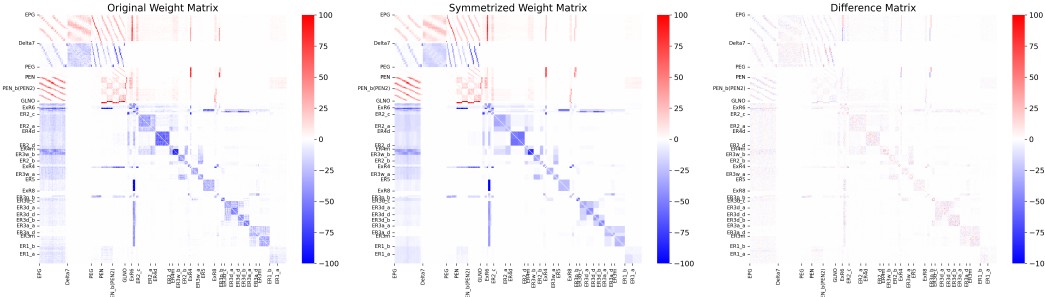

Figure A1: Weight matrix symmetrization.

All experiments we're run with weight symmetrization but eliminating weight symmetrization is possible if left and right populations of each cell type are treated as separate cell populations which can be tuned independently. This doubles the number of cell types but comparable results are achievable using this parameterization, even when weight symmetrization is not applied A2.

## A.2    Single parameter and E/I controls

In addition to the cell-type parameterization of the model, we also performed baseline experiments using a single global parameter to parameterize the synapse gains across all cell-types (thus not tuning differential gains between cell types), as well as using two parameters to describe the excitatory and inhibitory neuron gains separately. These reduced parameterizations failed to produce solutions capable of forming or integrating a bump even after model training, demonstrating the importance of cell-type parameterization (Figure A4).

## A.3    Kernel PCA and loss landscape visualization

In order to visualize the optimization landscape, we ran training runs with differently seeded initial parameters. We tracked the parameter values throughout training. We then used kernel PCA

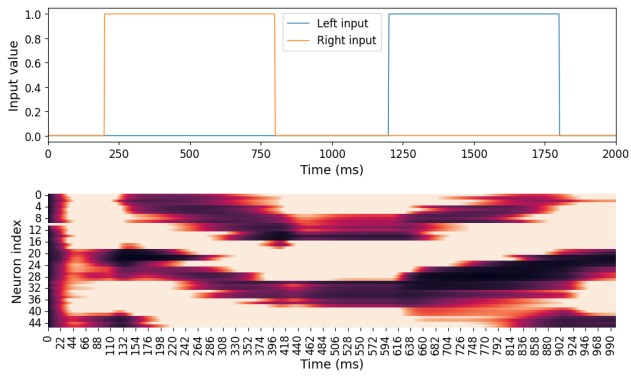

Figure A2: Neural dynamics after weight symmetrization is removed but left and right hemispheres are allowed to have different biophysical parameterizations.

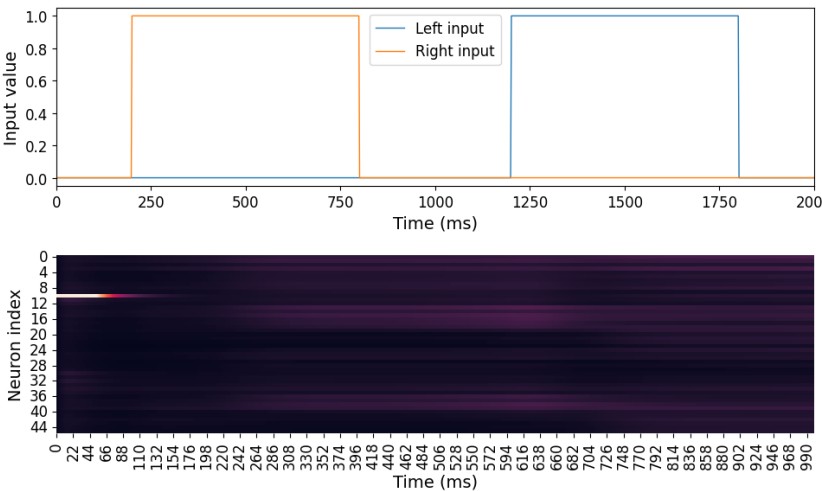

Figure A3: Visualization of the dynamics of an untrained network.

(Schölkopf et al., 1998) with a radial basis function kernel to fit the training trajectories and used a Kernel Ridge regression to learn the mapping from the basis to the original parameter space which we use to sample parameter values on in order to construct a two dimensional loss landscape.

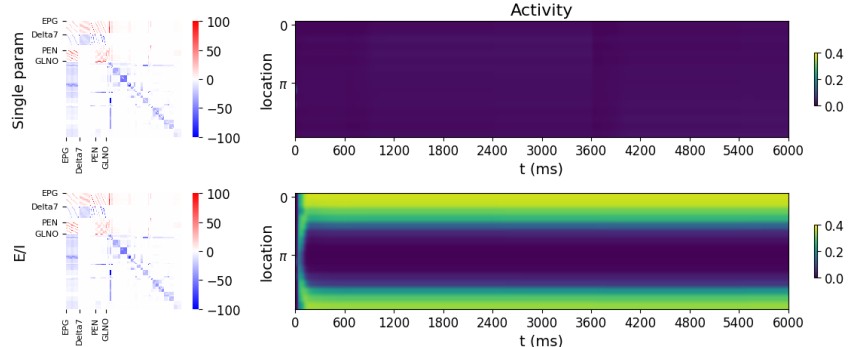

Figure A4: Baseline experiments. Learned weights and ring activity after optimizing only a single global gain parameter $w_0$ (top), or optimizing two parameters describing excitatory and inhibitory synaptic gains (bottom).

## A.4 Hyperparameter sweeps and model selection

For each ablation experiment (including the no-ablation condition), we trained models across a set of 90 hyperparameters by using a grid search over the initial bias values, leak, and global synapse strength. We additionally randomly initialized the block weights using a normal distribution centered at 1 with variance of 2 percent of the parameter value and swept over five seeds for each hyperparameter setting. In Figure 5 we chose the best 5 models from each condition to evaluate. We define best as the models with the smallest minimum speed loss (i.e., the models with the greatest range of velocity integration), after filtering for a R2 of > 0.7 and a temporal consistency loss of < 0.013.

## A.5 Delta7 ablations with full ring parameterization

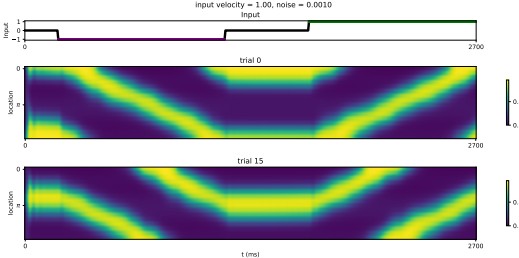

Figure A5: Activity bumps in Delta7 ablation, full ring parameterization.

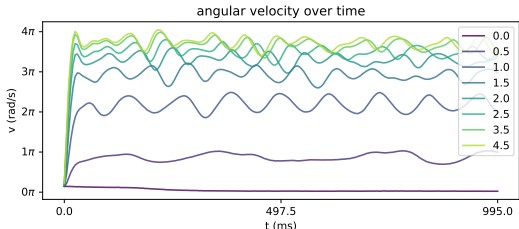

Figure A6: Internal vs. input velocity in Delta7 ablation, full ring parameterization.

## A.6 Sample weights from ablation experiments

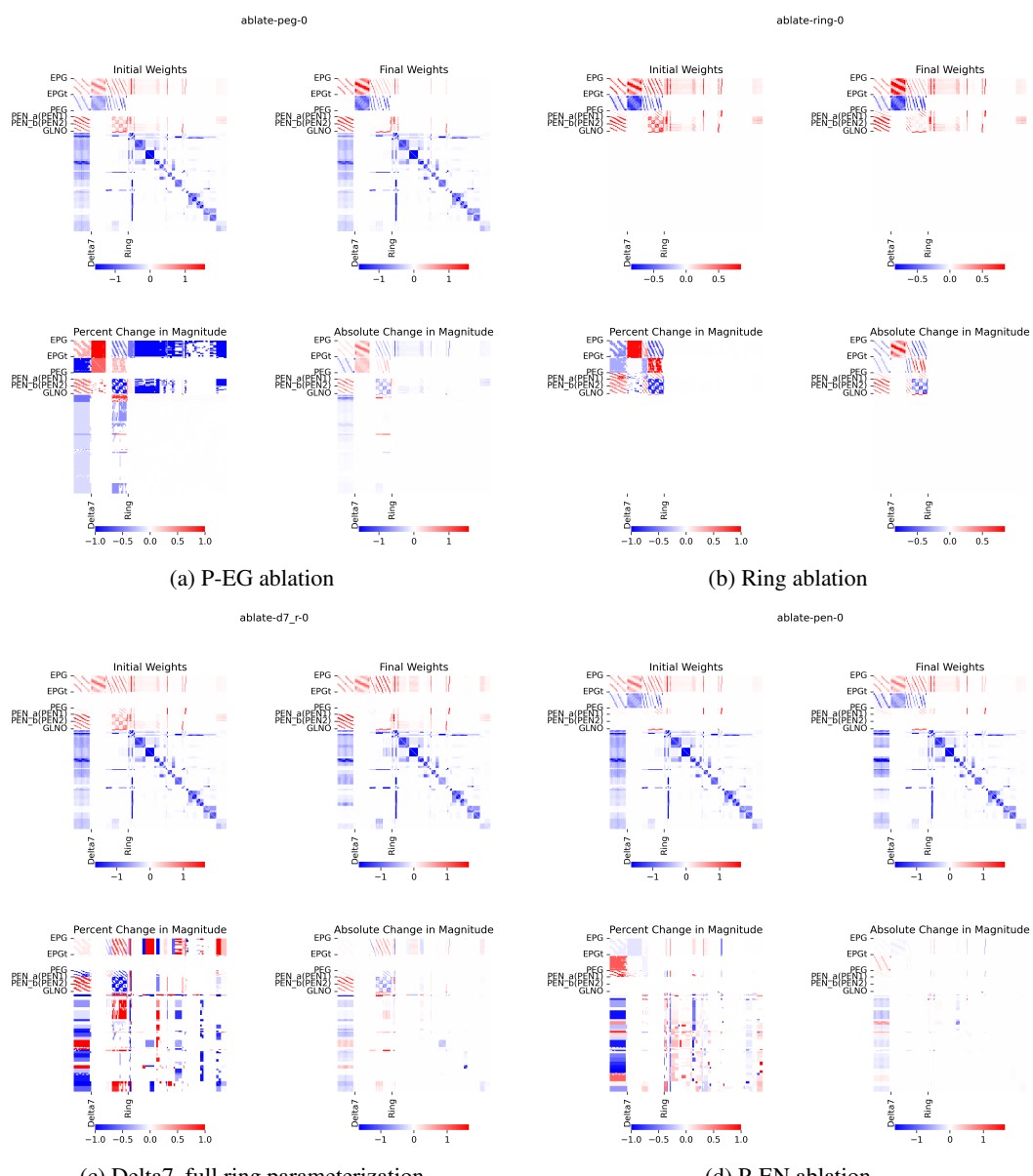

(a) P-EG ablation

(b) Ring ablation

(c) Delta7, full ring parameterization

(d) P-EN ablation

Figure A7: Ablation effects on trained synaptic weights: P-EG, Ring, Delta7, and P-EN.

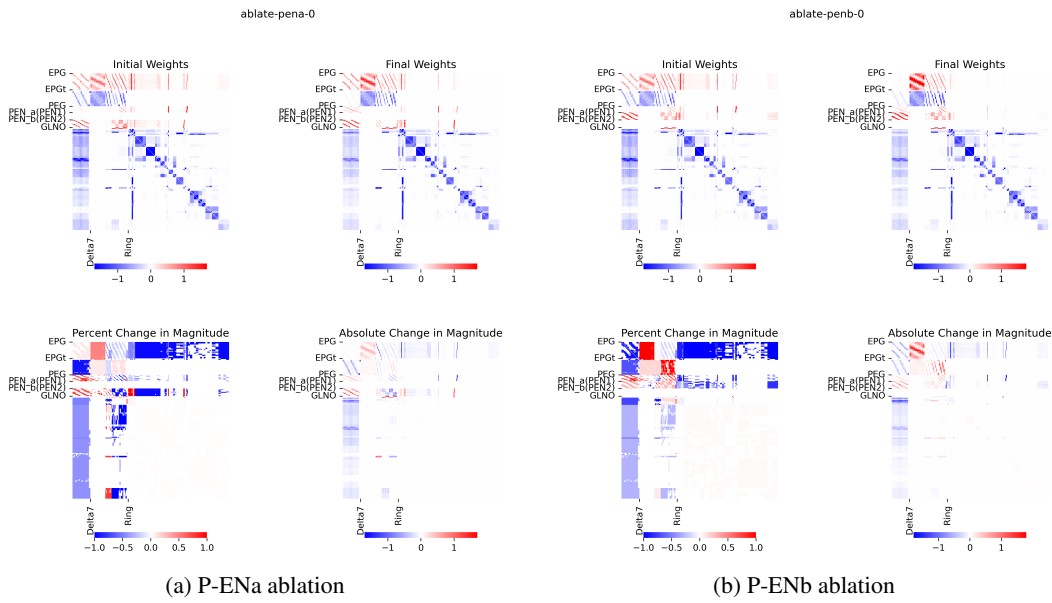

(a) P-ENa ablation        (b) P-ENb ablation

Figure A8: Ablation effects on trained synaptic weights: P-ENa and P-ENb.

## A.7 Additional Supplementary Methods

**Velocity input** Velocity input is driven by differential activity between the left and right GLNO neurons. In our experiments, we drove the velocity using either the left or right GLNO neurons at a given time. It is sufficient to drive differential activity between the left and right GLNO neurons to drive the velocity input which can be shown in A10.

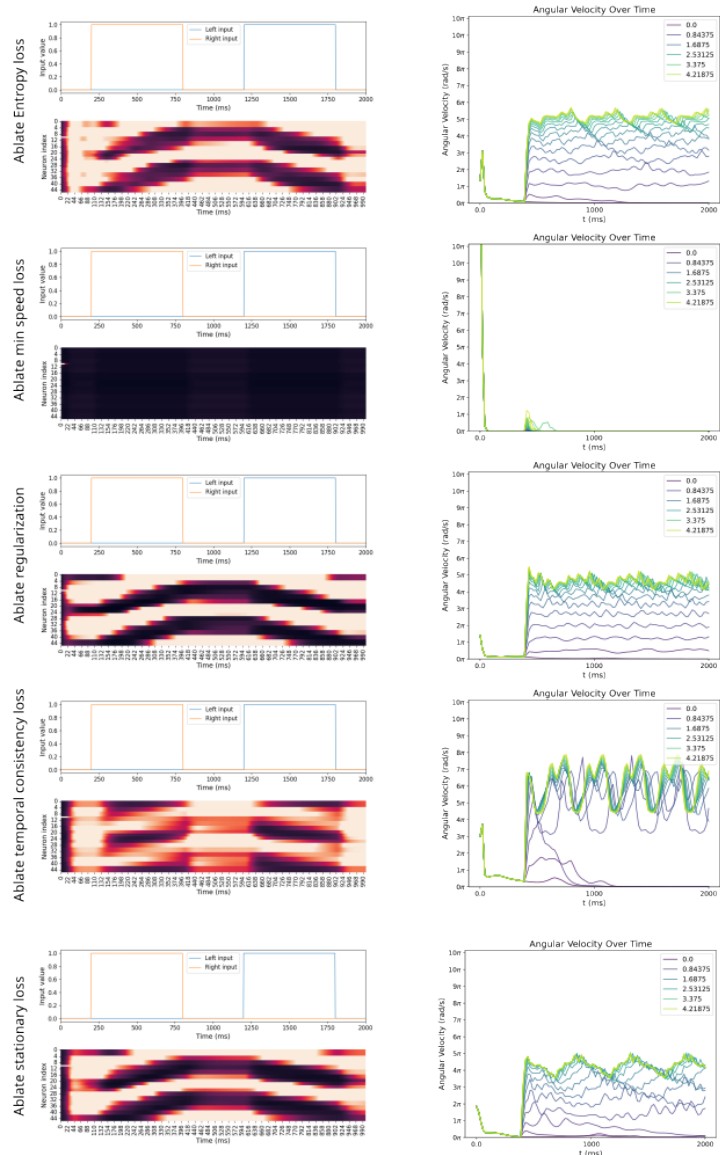

Figure A9: Neural Dynamics after ablating various loss terms. We show an example neural trajectory on the left hand side rotating in both directions as well as the angular velocity over time under different input conditions

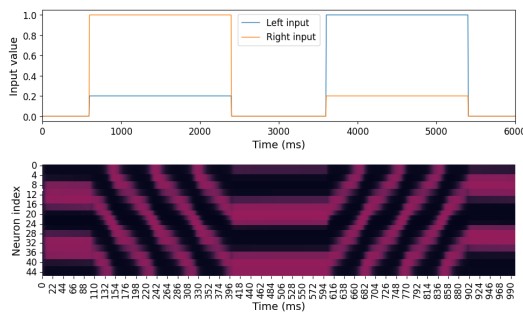

Figure A10: Activity trace when reversing sign of velocity input

## A.8 Supplementary Figures

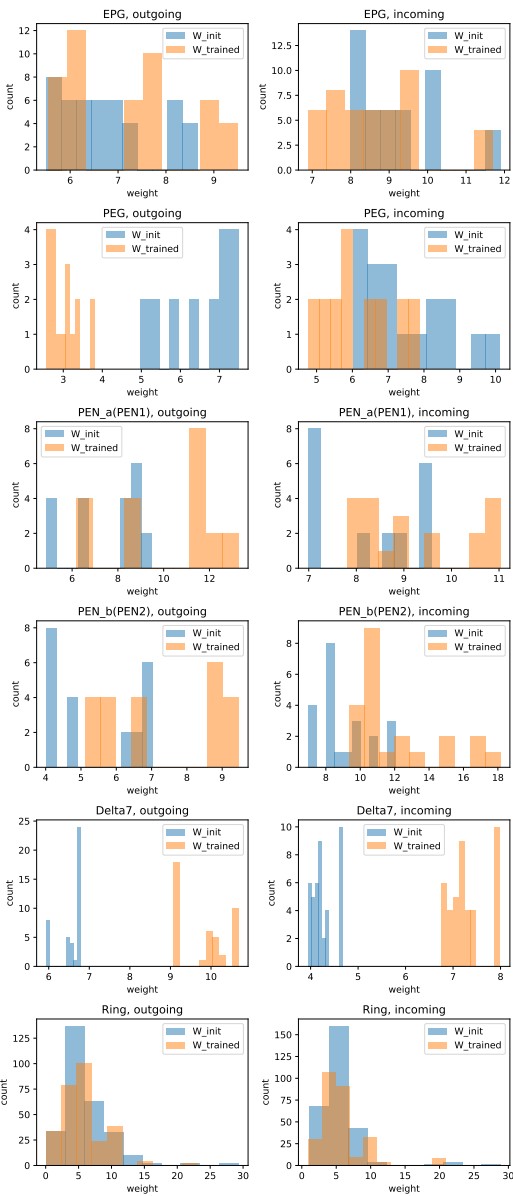

Figure A11: **Synapse statistics.** Distribution of incoming and outgoing synapse weights before and after training of a full connectome-initialized network.

