# OpenReview forum: "From Synapses to Dynamics: Obtaining Function from Structure in a Connectome Constrained Model of the Head Direction Circuit"
_NeurIPS.cc/2025/Conference — NeurIPS 2025 poster_

### Official Review · Reviewer_hYrM · 2025-06-05

**Clarity:** 4
**Significance:** 4
**Originality:** 4
**Rating:** 5
**Confidence:** 5

**Summary:**

Authors conduct a computational study on the head direction circuit in fruit fly by using the recently published connectome dataset. They train networks that are constrained on the connectome and show that a continuous attractor (which is often studied with hand-picked models in the literature) emerges from a self-supervised training. Overall, the authors provide several distinct insights into the neurobiology of the head circuit with a rigorous and well-executed study.

**Questions:**

I have the following suggestions:

- Could you please make it clear in lines 182-192, which of these loss functions are your unique contributions and which ones are used in the literature. I also think linear consistency loss needs a bit more clarification for someone who is not immediately expert (i.e., why does this incentivize linear changes wrt input)? For instance, most of the setup seems very similar to this seminal work: https://openreview.net/forum?id=8ox2vrQiTF

- Lines 159-160: I think you can also motivate this through gene expressions and more broadly biological motivations. For someone outside of neuroscience, it may not be clear why this is a smart choice.

- There could be a better explanation for why the linear consistency is not considered supervised learning. I think that might also be more appreciated by the NeurIPS audience.

- I do believe this study would be at a spotlight quality if there was an additional experiment with RELU or leaky-relu or selu or soft-plus nonlinearity. The choice of nonlinearity is arbitrary, as authors also point out. It would have been desirable to show the same results with a different nonlinearity, though I do not think this is absolutely necessary for acceptance.

**Ethical Concerns:**

["NO or VERY MINOR ethics concerns only"]

**Final Justification:**

As noted in my review, I think this work is an interesting addition to Neurips.

**Limitations:**

NA.

**Paper Formatting Concerns:**

NA.

**Quality:**

4

**Strengths And Weaknesses:**

The methodology is sound, the ablation studies are on point, the training process is exceptionally clever. I am having trouble finding things that are wrong with this work, see my final point below for suggestions for improvements.

---

> ### Author Rebuttal · Authors · 2025-07-31
>
> Thank you for your thoughtful review, as well as for your helpful suggestions that will help us clarify and improve our paper! Below, we respond explicitly to each of the specific points and questions you raised. Given the constraints of the rebuttal guidelines, we are unable to include figures and data. However, we have described additional analysis that we have performed that we would gladly incorporate into the paper's final revision:
>
>
> > Could you please make it clear in lines 182-192, which of these loss functions are your unique contributions and which ones are used in the literature. I also think linear consistency loss needs a bit more clarification for someone who is not immediately expert (i.e., why does this incentivize linear changes wrt input)? For instance, most of the setup seems very similar to this seminal work: https://openreview.net/forum?id=8ox2vrQiTF
> Thank you for the great suggestion; we will include a more detailed discussion of our loss terms in the paper.
>
> With regards to the linear consistency loss, the formula presented has a typo in it. The corrected loss term should read
> $\frac{1}{T} \frac{1}{2U} \sum_t^T \sum_{u=-U}^U \left( \frac{\delta \theta(t)}{|u|} - \mu\right)^2$ where $\mu = \frac{1}{T} \frac{1}{2U} \sum_t^T \sum_{u=-U}^U \frac{\delta \theta(t)}{|u|}$ which penalizes the variance of the ratio between changes in the internal state (decoded angle) and input velocity, ensuring that changes in internal state are proportional to changes in input drive. We will clarify this in the final manuscript.
> Likewise, thank you for bringing our attention to the connections between our work and the Schaeffer et al. (2023) paper, which also uses self-supervised learning to train a circuit to learn an internal representation of a spatial variable. We designed our loss terms independently based on biological constraints we believed should apply broadly across neural circuits. This has resulted in our linear consistency loss converging conceptually to the conformal isometry loss proposed by Schaeffer et al. (2023), both of which enforce that internal neural representations update proportionally with input magnitude. Specifically, we penalize the variance of the ratio between changes in the internal state (decoded angle) and input velocity, thus encouraging a consistent (linear) relationship of input velocity and velocity along the ring. However, there are clear distinctions between our approach and theirs: Schaeffer et al. (2023) impose explicit structural constraints (non-negativity, unit-norm) on their randomly initialized neural representations to learn Euclidean geometry directly. By contrast, we do not explicitly constrain neural activity; instead, our model starts from biologically measured connectomic structure, and implicitly learns biologically consistent representations through our self-supervised loss terms.
>
> Specifically, in Schaeffer et al. (2023), the authors train a randomly initialized MLP where “the neural representations are enforced to be non-negative and unit-norm via the non-linearity”. In contrast, we initialize our network from the connectome (thus constraining the connectivity based on animal measurements), and do not explicitly constrain the neural representations, instead asking the network to implicitly learn good representations through our loss terms (stability, entropy, and minimum-speed) which are formulated based on broadly accepted biological principles of neural coding, i.e., efficient, stable representations. Additionally, our losses apply more generically across a range of neural circuits, in contrast to the separation and capacity losses from Schaeffer et al. (2023) which are more specific to spatial representations. Overall, while we view our loss terms as distinct from those in Schaeffer et al. (2023), we now see the clear relevance of their work to our paper, and will add references to their work in our paper.
>
> > Lines 159-160: I think you can also motivate this through gene expressions and more broadly biological motivations. For someone outside of neuroscience, it may not be clear why this is a smart choice.
>
> Thank you for pointing this out. We will include more detailed discussion as to why parameterizing models at the cell-type level is motivated by biological evidence: neurons sharing cell types typically exhibit similar gene expression patterns, intrinsic electrophysiological properties, and connectivity motifs. These shared properties are thought to underpin similar functional roles within neural circuits. Cell-type-level parameterization thus aligns naturally with biological reality, reduces model complexity, and limits overfitting to individual neuron variability, while still capturing key functional diversity.
>
> > There could be a better explanation for why the linear consistency is not considered supervised learning. I think that might also be more appreciated by the NeurIPS audience.
>
> Thank you for this feedback! We consider our linear consistency loss to be self-supervised because it enforces internal consistency of neural representations without providing explicit external labels or targets for neural activity. Instead of prescribing specific activity patterns or target states, it constrains neural updates solely based on internal representational properties (e.g., proportionality to input velocity). Thus, the training data itself—the circuit’s own dynamics under various inputs—generates the supervision signal, making the approach self-supervised rather than externally supervised. We will edit the text with a better explanation of this.
>
> > I do believe this study would be at a spotlight quality if there was an additional experiment with RELU or leaky-relu or selu or soft-plus nonlinearity. The choice of nonlinearity is arbitrary, as authors also point out. It would have been desirable to show the same results with a different nonlinearity, though I do not think this is absolutely necessary for acceptance.
>
> Thank you for the great suggestion. We chose the sigmoid nonlinearity primarily because it closely resembles biological neural activation, which is characterized by saturating, bounded, and non-negative firing rates. We excluded non-linearities which are not lower-bounded and thus permit non-biological neural activity.
>
> That said, we have also been interested in exploring how alternative activation functions such as ReLU would perform, particularly given its simplicity in theoretical analyses and success in the machine learning literature. However, changing the nonlinearity is a nontrivial modification to the model (e.g., firing rates will no longer be upper-bounded) and will thus require additional care beyond simply swapping out the activation function. We plan to examine such alternatives in future work, and agree that demonstrating that our results are robust to choice of nonlinearity will greatly strengthen the generality of our method.

---

> ### Comment · Reviewer_hYrM · 2025-08-01
>
> As noted in my review, I think this is an interesting neuroscience work. I keep my stance that this work deserves acceptance.

---

> ### Comment · Reviewer_hYrM · 2025-08-06
>
> As authors may be aware, there is an active reviewer discussion happening in parallel. As part of this discussion, an important point was raised and I would like to hear authors' response on the resulting question.
>
> Could the authors provide a more principled and experimentally grounded roadmap for testing the predictions in figure 5? What can experimentalists do that will be a very clear causal test? Can you be more specific about underlying biology in general?
>
> Thanks!

---

> > ### Author Response · Authors · 2025-08-08
> >
> > Our predictions from our in-silico ablation experiments differentiate the contributions of distinct cell types, which can be tested experimentally via targeted optogenetic or chemogenetic silencing of each neuron type in a fly navigating a virtual reality arena, paired with measurements (e.g., via two-photon microscopy) of the animal’s internal head direction representation to compare changes in bump stability and angular velocity integration abilities. We outline specific causal experiments that can be used to test our model predictions below.
> >
> > For the Delta7 neurons, which we predict to be essential for bump formation, we expect this silencing to result in complete loss of circuit function by destabilizing and abolishing the bump. However, in our model, the network is able to recover its function under a relaxed cell-type assumption where it is allowed to separately tune the weights of the ring neuron subtypes. This suggests that selective plasticity in the ring neurons following Delta7 ablation can rescue network function by compensating for the missing inhibition. This could be tested by performing the experiment in animals during early development, when the circuit is more plastic, or by manually enhancing the activity of specific ring subtypes (e.g., using optogenetic stimulation).
> >
> > As we found that PEG neurons help stabilize the bump, we predict a more subtle effect: silencing PEG neurons should not eliminate the bump, but would significantly compromise its stability, particularly under noise. Concretely, we expect increased bump diffusion and less stable head direction estimates when visual cues are removed, or when controlled sensory noise is added in the environment.
> >
> > Finally, our simulations suggested differential roles of the PEN neuron subtypes, with the PENb ablation having a more detrimental effect. We can validate these distinct roles by showing a graded impairment, where silencing PENb (vs. PENa) neurons is predicted to cause both a relatively higher rate of spontaneous drift, which can be tested via its ability to maintain a stable head direction in the absence of visual cues, and a lower range of velocities at which the bump can accurately track visual motion, e.g., by using a rotating visual scene.

---

> ### Comment · Reviewer_hYrM · 2025-08-08
>
> Dear Authors,
>
> Thank you for the responses. I will admit, it is a bit underwhelming. A desirable one should have had new information, not restatement of claims we already read. For instance, what other cell types (maybe upstream?) are  good targets. Regardless, I will keep my score since I do not see any clear argument for a score increase.

---

> ### Comment · Reviewer_hYrM · 2025-08-08
>
> Sure, edited as requested. Although, I note clear disagreement. People do volunteer time, and thanking them and engaging with the argument is basic human decency. I dont get what is debatable here, but let's focus on the paper.

---

### Official Review · Reviewer_VhDz · 2025-06-12

**Clarity:** 2
**Significance:** 2
**Originality:** 2
**Rating:** 3
**Confidence:** 4

**Summary:**

This paper presents a computational framework to infer functional dynamics directly from a raw, synapse-level neural connectome, addressing the fundamental question of how anatomical structure specifies circuit function.  The authors use the Drosophila head direction (HD) circuit, a known continuous attractor network, as a testbed.  Their core contribution is a method that optimizes a small, biologically-interpretable set of parameters at the cell-type level (e.g., synaptic gains between cell types, neuronal biases) rather than for individual neurons or synapses.  The model is trained using a self-supervised objective that enforces linear velocity integration, without requiring any neural activity recordings or behavioral labels.  The resulting model successfully reproduces robust attractor dynamics, matching key experimental findings.  The authors then use this functional model to perform in silico ablation experiments, probing the specific roles of different cell types and generating new, testable hypotheses about the circuit's operation.

**Questions:**

On Connectome Preprocessing and Robustness: The paper states that the connectome data underwent a "symmetrization process" (§11.7).  Could the authors please elaborate on the specifics of this procedure in the main text or a supplement? How essential is this step to achieving the reported results? A demonstration that the model is not overly sensitive to this specific preprocessing choice would strengthen the paper. My evaluation would improve if the authors can show that the symmetrization is a minor, justified correction and that the model's success is not brittle with respect to this choice or the key hyperparameters.

On Generalizing the Self-Supervised Objective: The paper suggests the framework could be extended to other functions like "memory, or normalization".  This is an exciting prospect. Could the authors provide a concrete example of how the self-supervised loss function could be formulated for a different computation, for instance, divisive normalization, without relying on target activity patterns? Clarifying this would help the reader understand the true scope and generalizability of the proposed methodology beyond integrator circuits.

On the Role of P-EN Subpopulations: The finding that ablating the P-ENb subpopulation has a more detrimental effect than ablating P-ENa is a specific and novel prediction.  Can the authors leverage their trained model to provide a more mechanistic hypothesis for this difference? For instance, do the learned synaptic gain parameters (Z_AB) or the underlying connectivity counts (C_ij) reveal a systematic difference in how these two populations contribute to the circuit that explains the ablation result? Linking the emergent functional difference back to the structural or parameterized weights would provide a deeper insight.

On the Level of Abstraction (Cell-Type vs. Sub-Type): The finding that a Delta7-ablated network can be partially rescued by tuning the weights of individual ring neuron subtypes is fascinating and seems to run counter to the main thesis that cell-type level tuning is sufficient.  Does this suggest that the optimal level of parameterization is circuit-dependent? Could the authors comment on the principles that might guide a researcher in choosing the right level of abstraction (e.g., cell-type, subtype, or even single-neuron) when applying this framework to a new circuit?

**Ethical Concerns:**

["NO or VERY MINOR ethics concerns only"]

**Final Justification:**

I still believe that this work arrives at a common conclusion by applying a specific data analysis method to a particular dataset.

The method is not novel and there is no new insight from this work.

I keep my score.

**Limitations:**

Yes

**Quality:**

3

**Strengths And Weaknesses:**

Strengths:

Significance: The central idea of finding a "middle ground" of parameterization at the cell-type level is a powerful and important contribution.  It provides a principled approach to make connectomic data functional, moving beyond hand-tuned models without resorting to overly complex, unconstrained optimization. The self-supervised training paradigm, which obviates the need for large-scale neural recordings, is highly significant and enhances the scalability of this approach to other circuits where such data is unavailable.



Quality: The manuscript is of high technical quality. The authors do not just show that their model can integrate velocity; they perform a rigorous analysis of the learned dynamics using PCA, intrinsic dimensionality analysis, and persistent homology to demonstrate that the model has indeed learned a low-dimensional ring attractor manifold.  The ablation studies are particularly strong, as they involve retraining the ablated networks, providing a much more compelling test for the necessity of a given cell type than simply removing it from a pre-trained model.



Clarity: The paper is exceptionally well-written, clear, and logically structured. The motivation is well-established, the methods are described precisely, and the figures are clean, informative, and directly support the claims made in the text. Figure 1, in particular, provides an excellent and intuitive schematic of the core methodology.
Originality: While connectome-constrained modeling is an active area of research, the specific combination of self-supervised learning with a cell-type level parameterization applied directly to a raw, asymmetric connectome is novel.  The approach successfully recovers function from a biologically realistic (i.e., noisy and imperfect) structure, which is a key departure from models that assume idealized connectivity.


Weaknesses:

Dependence on A Priori Functional Knowledge: The self-supervised loss function is predicated on knowing that the circuit performs linear velocity integration.  This is a reasonable assumption for the well-studied HD system, but it limits the framework's applicability for true de novo discovery of function in less-understood circuits. The authors acknowledge this, but it remains a primary limitation on the generalizability of the current method.

Clarity on Preprocessing and Hyperparameters: The paper mentions a "symmetrization process" applied to the connectome synapse counts as a preprocessing step.  The details are deferred to an appendix, but this step could be critical. If the raw connectome requires significant "correction" to work, it slightly weakens the claim of obtaining function directly from the measured structure. Additionally, while the paper mentions that key global parameters were chosen via hyperparameter optimization, the sensitivity of the results to these choices, as well as the loss weights, is not explored. This leaves open the question of the model's robustness.

---

> ### Author Rebuttal · Authors · 2025-07-31
>
> Thank you for your thorough review! Your questions and suggestions highlight important points that we will make sure to address to improve the clarity and interpretability of our manuscript. Below, we provide detailed responses addressing each of your points. Given the constraints of the rebuttal guidelines, we are unable to include figures and data. However, we have described additional analysis that we have performed that we would gladly incorporate into the paper's final revision.
>
> > Dependence on A Priori Functional Knowledge…
>
>
> You raise an important point about the generalizability of our current loss formulation. Indeed, our linear consistency loss is motivated by the known integrative function of the Drosophila HD circuit. However, the critical underlying assumption—equivariant responses to input stimuli—is not circuit-specific and broadly applies to neural circuits that encode continuous variables (such as spatial position, head direction, or eye position). Thus, the overall framework we introduce does not inherently rely on detailed knowledge of neural representations or neuron-specific roles, and can broadly be applied to other integrator circuits in the brain. We are currently working towards extending our method beyond integrator circuits. For instance, we can remove assumptions about linearity and knowledge of the decoding function by redefining our linear consistency loss based on distances in the neuron activity space, rather than in the latent variable space, to enforce that larger changes to circuit inputs result in larger differences in population activity. This should work as long as the neural encoding of similar variables (i.e. nearby angles) are also similar, and we ensure that comparisons are within a reasonable range of inputs (resulting in internal representations where there is some overlap in neural activity). Nevertheless, we believe this present work, which focuses on integrator circuits, provides a valuable first step towards building more general models that leverage raw connectomics data to perform functional inference.
>
>
> > Clarity on Preprocessing and Hyperparameters…
>
>
> The symmetrization process we used does not impose any type of rotational symmetry on the matrix but rather leverages known symmetry between brain hemispheres which means it is relatively light and does not rely on any parameterization. Nonetheless, we have also performed experiments in which this symmetrization is removed but cell types are divided based on their spatial location in the left vs. right hemispheres. In this setup, the model is able to learn to integrate. We can include this experiment in our appendix as well to show that the symmetrization process is not essential to our results.
> We have observed that a number of different hyperparameters do support integration and can include additional hyperparameter choices in the appendix. We included a number of hyperparameter sweeps in order to eliminate the effect of the hyperparameters on the interpretation of the ablation results. We also believe that our results in Figure 5 also demonstrates the robustness of our model at least with regard to initialization as a number of different seeds are able to ultimately integrate well.
>
> > On Connectome Preprocessing and Robustness…
> Details about the symmetrization process are included in the appendix. Are there other aspects which require additional clarification? We have additional results which show that similar results are achievable without any symmetrization at all if cell types are divided based on their membership in the left vs. right hemispheres. The most notable effect of eliminating the symmetrization is that the shape of the bump is altered slightly since there is more heterogeneity within the sections of the ring. However, the network is able to integrate smoothly and has a linear velocity response as in the network presented in the paper.
> > On Generalizing the Self-Supervised Objective…
>
>
> Thank you for highlighting this! Indeed, although our current linear consistency loss explicitly focuses on integration by enforcing input-output equivariance, the underlying principle of our method—using self-supervised learning to enforce internal consistency of neural representations—can naturally generalize to other neural computations. As mentioned earlier, relaxing the linearity assumption could readily extend our approach to circuits implementing memory or other persistent representations of continuous variables. In the case of divisive normalization, one could employ a proportional scaling consistency loss that ensures relative neural response patterns remain stable across varying input intensities, while still preserving similar response patterns for similar inputs. Overall, we believe that training connectome constrained networks using minimal, biologically-grounded self-supervised constraints provides a promising framework for data-driven discovery (or confirmation) of neural circuit function from connectomics data.
>
>
> > On the Role of P-EN Subpopulations…
>
> Thank you for the great suggestion! To understand why P-ENb ablations were more detrimental than P-ENa ablations, we compared the learned synaptic strengths (recurrent, to/from EPG, and to/from GLNO; averages and variances were computed over 5 runs of each experiment) between the baseline and P-ENa/P-ENb ablation experiments.
>
> At baseline, P-ENa neurons receive stronger input from GLNO (average weight 1.81) compared to P-ENb neurons (1.35), and also have stronger connections onto EPG neurons (0.75 vs. 0.42 for P-ENb). Thus, initially, P-ENa forms the primary pathway from GLNO (input) to EPG (internal compass) neurons.
>
> However, when P-ENa neurons are removed, the network effectively compensates: GLNO-to-P-ENb connections strongly increase (from 1.35 to 2.28), P-ENb-to-EPG weights also increase (from 0.42 to 0.58), and EPG neurons boost their internal recurrence (from 0.06 to 0.16), restoring most of the circuit functionality.
>
> In contrast, when P-ENb neurons are removed, the network shows minimal compensation and does not appear to reroute signals. GLNO-to-P-ENa connections remain unchanged (1.81 baseline vs. 1.83 after ablation), and P-ENa-to-EPG connections also remain similar (0.75 baseline vs. 0.74 after ablation). EPG recurrence stays low (0.03).
>
> These findings suggest distinct and specialized roles for P-EN neuron subtypes: P-ENa neurons are the main pathway under normal conditions but are functionally redundant. In contrast, P-ENb neurons appear to fulfill a unique integrative role, perhaps due to specialized wiring or synaptic targeting, that make them uniquely essential for maintaining proper angular velocity integration.
>
> Thank you again for suggesting this analysis, and we will include more details in the supplementary information.
>
> > On the Level of Abstraction (Cell-Type vs. Sub-Type)...
>
> You raise an important point. We view this result as not as a contradiction but a refinement of our central claim: Our core method uses the broadest (cell-type-level) possible tuning to respect the granularity of current biological datasets and reduce overfitting. However, when key circuit elements are disrupted, introducing finer-grained tuning (e.g., ring neuron subtypes) can help restore function. This not only offers potential insights into the potentially redundant roles of different cell types and how they contribute to forming robust circuits, but may also illuminate important idiosyncrasies between cell (sub)types. Indeed, it is arguable that the ring neuron subtypes can each be considered their own cell type, as they each encode information about distinct aspects of the sensory environment, and synapse onto distinct dendritic locations on their postsynaptic neurons. This opens up the potential for a productive feedback loop, in which knowledge of cell types can be used for model training, and results from model training can refine cell type definitions in a hierarchical fashion.

---

> ### Comment · Reviewer_VhDz · 2025-08-03
>
> Dear Authors, I still have some questions:
>
> Turning the methods from Lappalainen et al. (2024) [1] into an unsupervised format is not novel; you should compare the advantages and disadvantages of the two methods with specific experimental results, rather than just demonstrating that a new method can run on a new dataset.
>
> The self-supervised modeling of neural dynamics based on single neurons or cell types' internal representational consistency has also been seen in previous research [2,3], and you should clarify how you differ from them, can these methods achieve similar function?
>
> [1] Lappalainen, J.K., Tschopp, F.D., Prakhya, S. et al. Connectome-constrained networks predict neural activity across the fly visual system. Nature 634, 1132–1140 (2024).
>
> [2] Learning Time-Invariant Representations for Individual Neurons from Population Dynamics. NeurIPS 2023
>
> [3] NetFormer: An interpretable model for recovering dynamical connectivity in neuronal population dynamics. ICLR 2025

---

> > ### Author Response · Authors · 2025-08-03
> >
> > Lappalainen et al. (2024) use a supervised optic-flow prediction task to train an idealized connectome that assumes perfectly periodic circuitry tiling the visual field; we utilize unsupervised learning on a minimally parameterized connectome-constrained network, which incorporates the asymmetries and irregularities (such as between sectors along the ring) present in the raw, measured connectome.
> >
> > Our work is complementary to that of Mi et al. (2023); their model trains on neural activity recordings to learn the dynamics of single neurons, which are then used to infer neuron identity. We do not use any activity recordings, and instead rely on the connectome - including cell class labels for each neuron -  to infer circuit-level dynamics and function.
> > Similarly, our work is fundamentally distinct from NetFormer (Lu et al., 2025), which attempts to infer connectivity structure from neural activity recordings. Both Mi et al. and NetFormer seek to extract structure from function; we do the opposite, deriving function from structure.
> >
> > The primary goal of our model is not to accurately predict dynamics on the level of single neurons, but to be able to characterize dynamics and function on the circuit level, using only connectomic data. None of the cited methods provide a suitable alternative for our method, as they cannot be used to infer network dynamics by leveraging only raw, non-idealized connectome measurements, including its inherent measurement noise and biological variation; Our method uniquely enables insight into the actual biological implementation of neural circuits, such as inferring the functions of specific cell types within a circuit, which allows for a better understanding of how real-world neural circuits achieve robust functional dynamics. This can be shown in our cell-type-specific ablation experiments – an analysis unique to our paper which we use to probe the unique functional roles of different neuron classes.

---

> > > ### Comment · Reviewer_VhDz · 2025-08-05
> > >
> > > Thank you for your rely! I still think it is not truly novel and insightful for me.
> > > I think it is a beautiful analysis work.

---

> > > > ### Comment · Reviewer_VhDz · 2025-08-07
> > > >
> > > > Dear All,
> > > >
> > > > I want to explain why i say like this.
> > > > I think this work analyzed a special dataset in a special way, and the analysis process was very good, so I say it analyzed beautifully, but did not generate new insights. It is like using an unconventional method to complete a regular proof problem. Its method is difficult to use for the analysis of other data and has limited impact.
> > > >
> > > > Thanks.

---

> > > > > ### Author Response · Authors · 2025-08-08
> > > > >
> > > > > Thank you for your continued engagement, we appreciate your openness. We would like to address some of the points that you are asserting to hopefully provide more clarity about our work.
> > > > >
> > > > > > analyzed a special dataset
> > > > >
> > > > > Our data comes from the Janaelia FlyEM project, a well-established and widely used connectome within the neuroscience community. While ongoing efforts like Flywire and MANC are collecting even richer, newer connectomes, extracting scientific insights remains a challenge. We believe our work is a step towards deriving insights from this data.
> > > > >
> > > > > > in a special way
> > > > >
> > > > > We would argue that producing biophysical simulations of working neural circuits is a long-standing technique and goal of many prior neuroscience projects. This is challenging to use on larger biological systems due to the amount of biophysical detail that is typically needed but with the advent of connectomic datasets, we believe that there is an opportunity to derive insights from simulation in this domain.
> > > > >
> > > > > > did not generate new insights
> > > > >
> > > > > We believe we have outlined our contribution in both demonstrating how to construct a working simulator of this circuit as well as utilize it to probe the function of various cell-types of the circuit, including some cell-types whose function is not immediately clear, such as the redundancy of the Delta7 inhibition and ring neuron inhibitions, the role of PENa/b neurons as well as the PEG neurons. While these predictions have not yet been tested experimentally, we believe that they demonstrate the utility of simulations for understanding neural systems.
> > > > >
> > > > > >difficult to use for the analysis of other data and has limited impact.
> > > > >
> > > > > While we admit that we cannot readily apply this method to other brain regions, this work is one of the first applications of simulation directly on connectomic data as well as the first of its type performed on an integrating system. The principles used to construct our system can be extended to other parts of the brain, and our parameterization based on cell-types can be immediately extended to other systems.

---

### Official Review · Reviewer_rEqo · 2025-06-26

**Clarity:** 3
**Significance:** 2
**Originality:** 3
**Rating:** 4
**Confidence:** 4

**Summary:**

In this work, the Authors propose a model to specify a circuit's function based on its known connectivity. To this end, they use a standard model for the dynamics of neural activity and learn its parameters (gain, bias, time constant) individually for each cell type to optimize a regularized task-specific loss. The model is then applied to the fruit fly's head direction circuit data from the FlyEM dataset where the Atuhors recover some known properties if the circuit (integrator dynamics; ring attractor network) and confirm/refine the roles of individual cell types in this circuit.

**Questions:**

Are all the terms of the loss function required to reproduce the results of this work? If so then, upon omission of terms, what results are not reproduced? If the results regarding the known properties of the neural circuit depend on the selection of the terms of the cost function, how can we be certain that the same wouldn't happen to the predictions regarding the unknown properties of the circuit?

How to extend this model to other circuits? How much prior knowledge of the circuit's function is required for the successful deployment of this model?

**Ethical Concerns:**

["NO or VERY MINOR ethics concerns only"]

**Final Justification:**

The Authors have clarified where they see the novelty/impact of the paper. Their responses have addressed some of my concerns.
The other concerns still remain, e.g., to further increase the impact of this work, one needs to *show* that the model can generalize to other circuits and to to just *suggest* that this might be the case.
Thus, based on the discussion with the Authors, I am raising the score by 1 point.

**Limitations:**

The loss function appears to require substantial information about the function of the neural circuit -- the property that the model has to predict.

My current scores on the paper reflect the uncertainty about how this cost function was formulated for the described task and how to formulate it for new tasks. I am happy to revisit my scores based on the interaction with the Authors.

**Quality:**

2

**Strengths And Weaknesses:**

Strengths:

The work deals with a highly important neuroscience problem of understanding the function/dynamics of a circuit bases on (incomplete, noisy) data describing its connectivity. Indeed, the connectivity data, typically obtained with the electron microscopy, while providing the information about the connectivity graph between the cells, lacks other important information including the strength of connectivity and other neuronal property. Additionally, the data is often noisy, featuring only the synaptic connections that were established with high confidence.

This work is also timely: as more high-volume high-resolution connectivity data becomes available for different species (worm, fruit fly, mouse), an opportunity present to study various circuits using these datasets, necessitating an algorithm that goes from the connectivity graph to a dynamical system. This paper presents such an algorithm.

The approximations/assumptions made in this paper are reasonable. For the neuronal dynamics, a standard equation is used. The parameters for this equation are unified within each cell type. These assumptions are then tested by recovering the known dynamics of the studied neural circuit and the known roles of its constituent cell types.

The results of the ablation studies are interesting and novel. Alongside with recovering the known properties of the studied neural circuit and its cells, the ablation studies here have helped to propose some novel properties of the circuit's cell subtypes.

The text is mostly well-written, well-structured and easy to follow.

Weaknesses.

The fruit fly's head direction circuit is a well-studied system, and substantial existing knowledge about this circuit seem to have been incorporated into the design of the loss function here. This includes the linear consistency, the stability, the minimum speed, and the entropy loss terms. I didn't see the ablation experiments for the components of the loss function but I assume that, since all of them are presented in the manuscript, all of them may be necessary to reproduce the known properties of the circuit in the proposed model. If that is true, then a successful model may require a substantial knowledge of the studied system -- something that it otherwise is supposed to provide. This property may also limit the application of the model to new, less studied systems.

For the loss function terms, I couldn't find the definition for capital theta.

---

> ### Author Rebuttal · Authors · 2025-07-31
>
> Thank you for your review and for highlighting both the timeliness and relevance of our work. We appreciate your feedback and hope to improve our paper by incorporating your suggestions. Below, we provide detailed responses addressing each of the specific points you raised.
>
> >The fruit fly's head direction circuit is a well-studied system, and substantial existing knowledge about this circuit seem to have been incorporated into the design of the loss function here…
>
> Thank you for bringing up this concern. We would like to clarify that while certain model components described in the paper are indeed tailored to the Drosophila HD circuit, the fundamental aspects of our approach are broadly applicable across other circuits with minimal modifications. We selected our loss functions based on general principles of neural computation that are broadly applicable across neural circuits, and believe our methods are readily generalizable to other circuits, particularly those that perform integration and representation of continuous variables.
>
> Specifically, the minimum speed and entropy loss terms were chosen to reflect the general principles of efficient neural coding: the entropy loss encourages evenly distributed activity across the population such that the network representation does not heavily depend on the activity of single neurons, while the minimum speed loss encourages the network to react to stimuli. Thus, we use these loss terms to prevent the trivial solutions of constant or zero activity. Likewise, the stability loss relies on the assumption that the internal representation should not change in the absence of perturbations, which is widely applicable to networks involved in memory and integration (we acknowledge that in circuits with spontaneous dynamics, this specific loss term may be modified or removed). Finally, the linear consistency loss enforces equivariant representations, i.e., that changes in internal state are proportional to changes in input drive. This assumes equivariant representations—a widely held principle in systems that encode continuous variables (e.g., head direction, spatial location, eye position). We agree that these latter two terms are most naturally suited to circuits that perform integration, and are currently working on extensions to our model to generalize beyond integrator circuits in the brain as well. Nonetheless, we believe that this work provides a valuable step towards building models that leverage raw connectomics data to perform inference of circuit function.
>
> We would also like to point out that while we do utilize some prior knowledge of network structure to compute our linear consistency loss, this knowledge is not required; our objectives can be easily generalized to systems where such prior knowledge is not available. Two concrete strategies to do this are:
>
> 1. decoder training: rather than assuming a known decoding function (e.g., population decoding of E-PGs), we can train a simple decoder (e.g., an MLP or linear readout) over a broader population to infer the encoded variable.
> 2. activity-similarity-based loss: even without a decoder, one can enforce that larger changes in the represented variable should produce more decorrelated activity across time, while small changes should produce similar activity patterns. This operates under the simple assumption the neural encoding of similar variables (i.e. nearby angles) are also similar.
> We are currently working to extend our model by testing these strategies in novel circuits with unknown or less-characterized functional roles.
>
> > Are all the terms of the loss function required to reproduce the results of this work? If so then, upon omission of terms, what results are not reproduced? If the results regarding the known properties of the neural circuit depend on the selection of the terms of the cost function, how can we be certain that the same wouldn't happen to the predictions regarding the unknown properties of the circuit?
>
> This is indeed an important point, and we will include results from and discussion of ablation experiments of each of the individual loss terms in the supplementary information. However, we reiterate that all of the loss terms we use either assume nontrivial encoding in neural circuits and can generically apply across all neural circuits, or assume equivariance, which holds for a wide range of circuits performing continuous variable integration. While our complete model is currently limited to integrator circuits, we see this work as an important initial step towards building more general models that can infer dynamics from the connectivity measurements of networks more broadly.
> I will briefly describe some results from our ablation experiments with individual loss terms here:
> * Linear consistency: The model is still able to integrate at a high velocity, but the integration velocity is not smooth nor linear.
> * Stability: The model is able to integrate, but there are fewer stable fixed points along the ring, resulting in less robust maintenance of heading direction.
> * Minimum speed: The model does not learn to integrate and collapses to a single fixed point.
> * Entropy: The model is able to achieve similar performance without this loss term and thus it is not strictly required; however, including it improves training stability and avoids activity saturation or collapse.
> * L1/L2 regularization: No major impact on integration ability; we include these primarily to promote biologically plausible (energy-efficient) neural activity
>
>
> > For the loss function terms, I couldn't find the definition for capital theta.
>
> Thank you for pointing this out! $\Theta(u)$ represents the Heaviside step function which is defined to be 0 when $u \leq 0$ and 1 when u is positive. We will edit our paper to include this.

---

> > ### Comment · Reviewer_rEqo · 2025-08-03
> > **Following up**
> >
> > Thank you for your response!
> >
> > While you touched upon all the concerns that I raised, it would be great to see the further support for your arguments so that I am convinced that they are indeed addressed. Please further comment along the following lines:
> >
> > - Based on your comments it looks like most if not all of the loss function components are necessary to reproduce the known properties of the Drosophila's head direction circuit. You further claim that these properties are universal in the optimal neural coding, however, I'd like to see the support for this claim. It would seem that, for example, in the hippocampus and in the olfactory system, there's an established representational drift - that goes against your stability part of the objective used here.  Or, while the entropy objective makes sense for maintaining the stability of the neural representations and such stability is reasonable as a hypothesis, I'd wonder whether it's as universal. For example, in the fruit fly, in the olfactory circuit, the cells on certain levels are highly specialized; also, in human, the so-called grandparent neurons were found, seemingly violating your proposal of that being a universal property of the optimal coding. Thus, it is important to see (1) the literature references supporting these claims and (2) for each claim, the scope where it applies.
> >
> > - I am excited to hear about the additional experiments that you are considering / conducting. Perhaps, hearing more about these could make your claims more convincing.
> >
> > I am looking forward to the Authors' response to these questions. Until then, I maintain my score.

---

> > > ### Author Response · Authors · 2025-08-05
> > >
> > > We would like to clarify that we do not claim all of our individual loss terms to be universal across all neural circuits. Rather, our statement was that they are broadly applicable across many circuits – particularly integrator circuits, which we focus on in this paper. In fact, these losses are not restricted to neural circuits but more generally applicable to dynamical systems which encode and integrate information.
> > >
> > > Linear consistency is a specific case of equivariance to input stimuli, an essential component to integrator circuits (Khona et al. 2022) since integration is inherently a linear operation. When inputs to the circuit are shifted or otherwise transformed, the circuit’s internal representation must shift in a corresponding manner in order to accurately represent and integrate input signals without distortion. This property is well studied in neural circuits that represent continuous variables such as spatial location (e.g., head direction cells and grid cells; see Burak and Fiete, 2009) and eye position (i.e., the oculomotor circuit; see Seung, 1996), and also applies to circuits encoding more abstract variables such as time (e.g., hippocampal time cells; see Kraus et al., 2013) and evidence (e.g., in the parietal cortex during decision-making tasks; see Shadlen and Newsome, 2001). Indeed, equivariance has been proposed to underpin generalization in not only spatial but also more abstract cognitive reasoning (Behrens 2018).
> > >
> > > We believe that there might be a misunderstanding regarding the entropy loss. Our entropy loss is computed across neurons, not over time, and thus favors sparse neural responses. Thus, the examples you mention of specialized cells – which minimize information redundancy by maximizing entropy – are in support of this sparse coding hypothesis. Entropy is a prevalent theme in neuroscience (Olshausen and Field 1996), and we enforce a relatively weak and biologically realistic constraint that simply encourages efficient and sparse activation patterns.
> > >
> > > Our stability loss is not specific to biological networks but is motivated by a general property of integrating systems in which internal representations, in the absence of input stimuli, must remain stable in order to accurately track the variable of interest. In the oculomotor circuit (see Seung, 1996), for example, deficiencies in stability lead to conditions such as nystagmus. While it is true that certain neural circuits – such as the hippocampal circuit – exhibit representational drift over longer timescales, this does not preclude the need for these circuits to exhibit local stability over short timescales, in order to allow downstream circuits to decode the variables they maintain.
> > >
> > > Our minimum speed loss, which prevents trivial solutions (constant zero activity), is likewise broadly applicable to all networks that need to react to external stimuli.
> > >
> > > As we previously discussed, the stability, L1/L2, and entropy losses are not strictly necessary to train our model, though they do promote better behavior in our tuned system. Likewise, we acknowledge that some of the loss terms we use are specific to a subset of neural circuits.  While this discussion about loss terms and generalizing across neural circuits is important and a future direction we are interested in and actively pursuing (specifically, we will be applying our model in circuits downstream of the HD circuit, in order to study multisensory integration, as well as in the visual circuit, to study stereoscopic vision), we would like to emphasize that we are not proposing a universal method for determining circuit function from structure, nor is it the primary focus of this work. Our intention is not to propose a universal set of specific loss terms, but rather demonstrate the use of an unsupervised framework that can be flexibly adapted to infer functional dynamics from structural connectivity measurements.
> > >
> > > Specifically, we see the primary contributions of our paper as demonstrating that biological connectomes contain rich information about circuit function, and introducing a minimally-parameterized connectome-constrained model capable of extracting such information to reveal mechanistic insight into biological circuit computations. Our model allows us to conduct in-silico experiments to probe circuit mechanisms in ways that would otherwise be infeasible in-vivo, providing a powerful tool for understanding biological circuits. To our knowledge, this is the first work to introduce a non-idealized connectome-constrained model of a non-trivially recurrent neural circuit, and demonstrate its use to make predictions about the functional roles of distinct cell-types within the circuit.

---

> > > > ### Comment · Reviewer_rEqo · 2025-08-05
> > > > **Re**
> > > >
> > > > Thanks for your clarifications. I am now engaging in the discussion with y'all and with the other reviewers to hopefully clarify the stated goal/merit/contribution of the paper.
> > > >
> > > > Your literature references are appreciated. Would it be correct to say that the components of the loss function that you use here can be split into general modeling assumptions that likely apply to all circuits (e.g. the equivariance) and the others specific to the continuous-variable/integrator circuits (e.g. the linear consistency)? Also, your point re: the entropy is taken.
> > > >
> > > > My remaining concerns are as follows. If the main claim of the paper is that *circuit function can be inferred from the connectome* -- it seems trivial and has been done time and again, so you are probably not doing that. If your claim is that you present a *method* to infer the function from the connectome -- its generality has to be shown. If you present a method and claim that it works for *integrator" circuits -- this claim would be substantiated by the text here, however, if for a certain circuit it's known that it's an integrator circuit -- what else is there to learn? Finally, if your claim is that you've *learned something new about the Drosophila HD circuit* -- that'd be highly valuable but then it need to be outlined and paraded.
> > > >
> > > > Which one is that? I would appreciate if you could let me know, as concrete as possible, what your claims are here. That'd help me to provide the most informed reviewer's report and will help the readers to contextualize your findings.
> > > >
> > > > Thanks! I'll stay tuned

---

> > > > > ### Author Response · Authors · 2025-08-05
> > > > >
> > > > > Our linear consistency loss is the component of the loss that enforces equivariance, but your interpretation is correct: some components of our loss enforce general principles of neural circuits, while others (like linear consistency and stability) are more specifically tailored towards integrator circuits.
> > > > >
> > > > > We present a method that infers circuit function from structural connectivity measurements that can be applied across integrator circuits, and apply it to learn novel insights about the Drosophila HD circuit.. For instance, our approach allows us to probe the different roles between the diverse set of cell types in the biological circuit (in contrast to the minimal types required by theoretical HD circuits). We appreciate your point that this should be highlighted more in the paper - we will make sure to do that.
> > > > >
> > > > > While it is true that our model, in its current implementation, can only be directly applied out-of-the-box to integrator circuits, we would like to note that it is a flexible framework that uses minimal prior knowledge; generalization to other circuits would require only a simple substitution of the linear consistency (and potentially stability) losses, in contrast to other connectome-constrained methods which require more knowledge and assumptions about the circuit architecture and task. To push back a little on the statement that circuit function can be trivially inferred from the connectome: to our knowledge, we are the first to construct a working connectome-constrained model of a recurrent, non-sensory circuit that is truly data-driven, enabling direct inference about the biological circuit implementation that is not possible with idealized models which impose structural priors. Though previous work exists in inferring function from network structure, there is a substantial gap between abstract inference and building a virtual twin that enables direct simulation and detailed analysis of a biological circuit, as we have done here.
> > > > >
> > > > > Finally, even if we know that a circuit functions as an integrator, there are many things we can still learn about the circuit! For example, we can gain insight into how biological circuits implement robust attractor dynamics despite structural asymmetries and noise present in nature, in contrast to the finely tuned connectivity required by theoretical attractor models (we discuss this in more detail in our response to reviewer 4Q4m). Additionally, although certain neural circuits are believed to implement low-dimensional attractors, there is little or no direct evidence that do not actually encode additional latent variables for many of these circuits; we show that the dynamics of the Drosophila HD circuit are indeed strictly 1D. This knowledge, along with previous work demonstrating 2D dynamics of the grid cell circuit in mammals, are crucial to better understanding if and how animals repurpose low-dimensional circuits supporting spatial navigation to generalize across many (non-spatial) tasks, as per the cognitive map hypothesis (see Behrens, 2018). Lastly, understanding the roles of distinct cell types through our method will allow us to better design perturbational studies to directly probe the circuit experimentally.
> > > > >
> > > > > Thanks for your continued interest and engagement!

---

> > > > > > ### Comment · Reviewer_rEqo · 2025-08-06
> > > > > > **TY**
> > > > > >
> > > > > > Dear Authors,
> > > > > >
> > > > > > thanks for your on-point response.
> > > > > >
> > > > > > All three major points here are convincing.
> > > > > >
> > > > > > While I still think that there are limitations to this work (e.g. I don't think that extending your model to other circuits would be as simple as you suggest; otherwise, that needs to be shown), in light of your responses here, I'd like to raise the score.
> > > > > >
> > > > > > Please edit the text though to frame the work and its significance along the lines discussed here, especially in your latest response. That will help the readers to contextualize your findings.
> > > > > >
> > > > > > I'm also here if you'd like to further discuss.
> > > > > >
> > > > > > All the best

---

> > > > > > > ### Author Response · Authors · 2025-08-06
> > > > > > >
> > > > > > > Thanks for your engaged discussion! We've received a lot of helpful and actionable feedback throughout this review process, and we'll make sure to incorporate them into our paper to better frame and contextualize our work for readers.
> > > > > > >
> > > > > > > As we are working to extend our model to other circuits, we hope you'll stay tuned for our future work :)

---

### Official Review · Reviewer_4Q4m · 2025-06-27

**Clarity:** 4
**Significance:** 3
**Originality:** 3
**Rating:** 5
**Confidence:** 3

**Summary:**

The authors address the challenge of deriving functional network dynamics from incomplete and noisy connectomic data of the Drosophila head direction (HD) circuit. They propose a method to infer missing neuronal parameters—such as thresholds and time constants—and to correct potential errors in synaptic connectivity. The resulting model accurately integrates angular velocity inputs and exhibits continuous attractor dynamics. Importantly, the authors demonstrate that tuning parameters at the cell-type level suffices to achieve the desired dynamics. They also perform in silico ablation experiments to elucidate the functional roles of specific neuron types.

**Questions:**

- Can the authors elaborate on the "noisy connectome measurements" to help us better understand the problems? What are the potential sources of measurement errors? How likely are they to appear, and what will be the consequences?
- What would the results be like if $Z$ and, therefore, synaptic connectivity are not tuned at all?
- Much prior domain knowledge is used in the design of the method, which limits its application to other circuits. To explore its full potential, I wonder how the method would behave if all of this knowledge were removed. By domain knowledge, I mean 1. external inputs are only injected to left/ right GLNO neurons; 2. a single E-PG neuron's activity is initialized to 1 while others are set to 0; 3. location is decoded from the activity of E-PG neurons.
- Can you explain in more detail the "population coding" you used to decode the location?
- Can you elaborate on how the hyperparameters are optimized?
- Can you elaborate on why the linear consistency loss encourages a *linear* readout of the velocity input?
- The meaning of $\Theta(\cdot)$ used in the loss function is unexplained.
- Can you elaborate on why $C_{ij}$ is used in L1 and L2 regularization? Are small synapse counts more subject to measurement errors?

**Ethical Concerns:**

["NO or VERY MINOR ethics concerns only"]

**Final Justification:**

The authors address the challenge of inferring functional network dynamics from incomplete and noisy connectomic data of the Drosophila head direction (HD) circuit. The proposed self-supervised learning approach, guided by biologically motivated training objectives, is both interesting and promising. The method also appears readily applicable—or adaptable with minimal modifications—to other integrator circuits.
Additionally, the results suggest that the connectome already contains substantial information about circuit function, as parameter tuning at the cell-type level proves sufficient.
Overall, I find the paper to be well-executed and complete with respect to the problem it seeks to address, and I recommend its acceptance to NeurIPS.

**Limitations:**

- My primary concern lies in the generalizability of the proposed method. The approach—particularly the training objectives and input-output relationships—appears tailored specifically to the Drosophila head direction circuit. A broader discussion of the scope and limitations of this generalization would significantly strengthen the paper.

- The authors argue that replicating idealized continuous attractor dynamics requires finely tuned connectivity with perfect rotational symmetry, which is not observed in the connectome. While this claim is common in theoretical models, I am not fully convinced that such symmetry and fine-tuning are strictly necessary in biological implementations. Dynamical systems with slow manifolds and discrete fixed points can exhibit ring-attractor-like behavior without strict rotational symmetry or extensive tuning. It is plausible that the Drosophila HD circuit leverages such robustness.
This raises an important question: how challenging is the problem if the system can function without the degree of tuning the authors assume? For example, what do the dynamics look like with the initial connectivity matrix and neuronal parameters? Including a comparison between this baseline model and the final optimized model would help clarify the extent to which the proposed optimization is necessary and what specific gains it affords.

**Quality:**

4

**Strengths And Weaknesses:**

Overall, I found the question interesting, the analysis solid, the results convincing, and the paper well-written.
While I am generally positive about the contribution, I outline some high-level concerns in the "Limitations" section below.

---

> ### Author Rebuttal · Authors · 2025-07-31
>
> Thank you for your thorough and thoughtful review, and constructive feedback! We have responded below to your specific concerns and questions.
>
> > Can the authors elaborate on the "noisy connectome measurements" to help us better understand the problems? What are the potential sources of measurement errors? How likely are they to appear, and what will be the consequences?
>
> In the paper, we provide a brief description of the noise in lines 30-31; we will expand our description to make the sources and consequences of connectome noise more clear. There are three primary sources for noise: connectomics data is primarily collected through electron microscopy, and errors can be introduced at multiple points throughout the process including image alignment of 2D slices, automatic detection of neuronal structures such as synapses, and manual proofreading. From our discussions with experimental labs, we have been made aware that human proofreading is a crucial but labor-intensive part of this process, and the amount of noise in connectome measurements varies depending on the level of proofreading. On the biological side, specific synapse numbers are also known to vary between animals and change across time, as animals learn and adapt in their environments. Thus, connectomic snapshots offer noisy measurements of biological networks, and require some level of parameter tuning to achieve a more accurate picture of network function.
>
> > What would the results be like if Z and, therefore, synaptic connectivity are not tuned at all?
>
> We have actually run baseline experiments that vary the level of tuning to demonstrate the importance of cell-type parameterization. We briefly mention this in Section 5.3, but will expand upon it and also include more details in the SI. In short, tying parameters globally across cell types does not result in a solution that forms or integrates the bump, regardless of whether the synaptic connectivity gain is tuned or not (i.e., if Z=1).
>
> > Much prior domain knowledge is used in the design of the method, which limits its application to other circuits...
>
> We have found that our method is robust to each of the specific assumptions you highlighted:
> 1. We used GLNO inputs to make our model more aligned with the biological circuit, but we have also previously tried injecting inputs through the PEN_a/b neurons as well as the E-PG neurons directly, and all work equally well. In fact, any source of initial net excitatory activity and time for the network to stabilize is sufficient to form a bump. It’s true that asymmetric inputs are needed for the bump to integrate, but we believe that local drive is a general assumption that applies broadly across the brain (e.g., retinotopy in vision).
> 2. Similar to 1, we chose local E-PG initializations to make our model more biologically aligned, but any source of symmetry breaking initialization will stabilize to a single bump through recurrent network interactions; we can generalize this by initializing neuron activities randomly instead.
>
> 3. Although we make use of the prior knowledge that E-PG neurons are known as the canonical ring neurons in the Drosophila brain, a more generic application of this approach could instead train a simple decoder (e.g., a MLP) to decode activity from a broader population of neurons involving more cell types. Alternatively, we can redefine our linear consistency loss by measuring change in the neuron activity space, rather than in the encoded variable.
> More broadly, we are also working to reformulate our model, including our loss terms, so that our method can be applied more generally to circuits beyond integrator networks in the brain.
>
> >Can you explain in more detail the "population coding" you used to decode the location?
>
> Population coding is a common neural coding strategy where information, such as head direction, is represented by the combined activity of a group of neurons, rather than a single neuron.
> As defined in Bialek et. al. 1989, population coding assigns a preferred direction (the direction they maximally fire at) to each of the neurons in the population, such that the direction can be decoded as the sum of the neurons’ directions weighted by their activity. We will expand our definition in our paper to clarify this.
>
> > Can you elaborate on how the hyperparameters are optimized?
>
> We describe the details of the hyperparameter search in the appendix, and will ensure this is referenced clearly in the main text. We performed a grid search over initial bias values, leak, and global synapse strength. We selected the best performing model based on the linearity of the model’s response to velocity stimuli.
>
> > Can you elaborate on why the linear consistency loss encourages a linear readout of the velocity input?
>
> Thank you for pointing this out. The formula presented has a typo in it. The corrected loss term should read
> $ \frac{1}{T} \frac{1}{2U} \sum_t^T \sum_{u=-U}^U \left( \frac{\delta \theta(t)}{|u|} - \mu\right)^2 $ where $ \mu = \frac{1}{T} \frac{1}{2U} \sum_t^T \sum_{u=-U}^U \frac{\delta \theta(t)}{|u|} $ which penalizes the variance of the ratio between changes in the internal state (decoded angle) and input velocity, ensuring that changes in internal state are proportional to changes in input drive. We will clarify this in the final manuscript.
>
> > The meaning of $\Theta ( \cdot )$ used in the loss function is unexplained.
>
> Thank you for pointing out this oversight. $\Theta(u)$ is the heaviside step function which is defined to be 0 when \(u \leq 0\) and 1 when u is positive; we will update the manuscript to include this.
>
> > Can you elaborate on why $C_{ij}$ is used in L1 and L2 regularization...
>
> The $C_{ij}$ are used since the final synapse strength is the product of $Z_{ij}C_{ij}$, thus scaling the regularization to be proportional to the synapse strength itself. If we use the gains $Z_{ij}$ instead, then the regularization would be relatively stronger on the smaller weights. This is also a valid design choice, but we believe it intuitively makes more sense to penalize deviations from the connectome on the scale in which these deviations affect the dynamics, i.e. the scale of synaptic counts.
>
> >My primary concern lies in the generalizability of the proposed method...
>
> You highlight a valid concern, and we will make sure to include a more detailed discussion about model generalization and limitations in our paper. We see the model described in the paper as an important step towards utilizing connectomes to characterize neural computations, and believe that it can be generalized naturally to a range of neural circuits for which the simple assumption of equivariant responses to inputs is applicable (as is known to be the case for many sensory circuits). As we mentioned earlier in our response, although we leveraged prior knowledge of population encoding of head direction in E-PGs in this paper, we can generalize this to other systems with minimal modifications by instead training an additional decoder on a broader set of neurons, and/or using a simpler approach based on measuring distances in the space of neural activity rather than a latent variable. In fact, we are now actively working to use these approaches to apply our method on novel circuits in the brain in which we do not know the precise roles of specific neuronal subtypes, or even circuit function.
>
> We agree that finding an even more general, unsupervised approach that can relax even the assumption of equivariant representations would generalize our method even further to a larger set of networks. We see this as an important future direction that we are building towards, starting with our current model, which contributes to the field of connectome-constrained modeling by introducing a novel approach to understanding circuit function from structure.
>
> > The authors argue that replicating idealized continuous attractor dynamics requires finely tuned connectivity with perfect rotational symmetry, which is not observed in the connectome...
>
> Your point about slow manifolds and finely tuned connectivity is apt and in fact, is in line with our model! Our model does not assume or enforce rotational symmetry, nor does it achieve (perfect) rotational symmetry through tuning. In many ways we are arguing the same point that you have raised: that despite some notable asymmetries (such as the variable number of neurons between each of the subsections of the ring) in the biological HD network, our minimal tuning is sufficient to produce a circuit that is still able to integrate. Our tuning process cannot correct for cell-level asymmetries, as we only adjust biophysical parameters on a cell-type level. This contrasts to previous works which build in assumptions about rotational symmetry in the network architecture, and cannot be used to make statements about the actual biological implementations of the network. Thus, we agree that it is very likely that biological circuits do employ slow manifolds and discrete fixed points without precise circuit symmetries in order to produce attractor-like behavior, and we believe that our method, applied to a circuit known for implementing attractor dynamics, is an important step in showing this is the case!
>
> > This raises an important question: how challenging is the problem if the system can function without the degree of tuning the authors assume?
>
> We have performed controls that we can include in the supplementary information, where we have compared the results of untrained systems as well as systems that have been trained with fewer parameters. We find that the amount of noise in the data as well as our lack of complete knowledge of the biophysical parameters of the system necessitate additional tuning.

---

> > ### Comment · Reviewer_4Q4m · 2025-08-03
> >
> > I thank the authors for their thorough response, which clarified many of my questions. I have a few follow-up points:
> >
> > - Could you provide more details on how the method performs when the "population coding" component is replaced with a simple MLP that decodes from the full set of neurons? This would help assess the generalizability of the proposed approach.
> >
> > - You mention that a range of neural circuits exhibit equivariant responses to inputs. As I am not deeply familiar with this area, it would be helpful if you could elaborate—specifically, what are some additional examples of such circuits where this assumption is known to hold? This would help clarify the relevance and broader applicability of the proposed method.

---

> > > ### Author Response · Authors · 2025-08-05
> > >
> > > Given the limited time of the discussion phase, we will not be able to report the results of training a MLP decoder within our model for this response. However, we would like to draw your attention to two papers that demonstrate the feasibility of the two alternatives we suggested as substitutes for population decoding: Lappalainen et al. (2023) train a two-layer convolutional decoder end-to-end with their connectome-constrained network of the optic lobe, while Schaeffer et al. (2023) define losses in the neural activity space instead of the latent variable space to train a grid-cell network using self-supervised losses. We discuss differences between these approaches and ours in the manuscript as well as in our response to reviewer hYrM.
> > >
> > > Regarding your second point: equivariance to inputs is an essential property of integrator circuits (Khona et al. 2022) because integration itself is a linear operation. When inputs to the circuit are shifted or otherwise transformed, the circuit’s internal representation must shift in a corresponding manner in order to accurately represent and integrate input (i.e. sensory/motor) signals without distortion. This property is well studied in neural circuits that represent continuous variables such as spatial location (e.g., head direction cells and grid cells; see Burak and Fiete, 2009) and eye position (i.e., the oculomotor circuit; see Seung, 1996), and also applies to circuits encoding more abstract variables such as time (e.g., hippocampal time cells; see Kraus et al., 2013) and evidence (e.g., in the parietal cortex during decision-making tasks; see Shadlen and Newsome, 2001). Indeed, equivariance has been proposed to underpin generalization in not only spatial but also more abstract cognitive reasoning (Behrens 2018).

---

> > > > ### Comment · Reviewer_4Q4m · 2025-08-05
> > > >
> > > > I thank the authors for their additional clarification, which has addressed my remaining concerns. I have no further questions and will raise my score to 5.

---

### Note · Authors · 2025-08-14

We are grateful to the reviewers for their thorough engagement and constructive feedback. The discussion has been highly valuable, helping us identify several avenues for improving the clarity and contextualization of our work, which we will incorporate into the final manuscript.

We believe the paper makes a notable contribution in its present form. Our study introduces a framework for deriving circuit function from raw structural data, demonstrating that a non-idealized biological connectome, with its inherent asymmetries and noise, is sufficient to produce robust attractor dynamics. By applying self-supervised learning objectives with minimal, cell-type-level parameterization, our model recovers the integrative function of the Drosophila head direction circuit directly from its measured wiring diagram. This approach, in turn, allows for in-silico experiments that yield novel, experimentally testable hypotheses regarding the functional roles of distinct neuron subtypes.

The reviewers have also raised thoughtful points about potential extensions and broader applications of this work. While these represent compelling directions for future investigation, we are cautious against overstating potential extensions without corresponding empirical results to ground our conclusions. We believe that the results presented in this paper as well as our approach represents a useful step in bridging the gap between connectomics and systems neuroscience, providing a data-driven tool for understanding how function emerges from complex neural structures.

---

### Decision · Program_Chairs · 2025-09-17

**Decision:**

Accept (poster)

**Comment:**

The goal of this work is to use recently collected data on the fly connectome to construct a working model of the fly head direction circuit based solely on connectivity data augmented with theoretically-motivated learning objectives. While a small number of previous papers have attempted the same structure-to-function task, these efforts have been in early sensory systems with primarily feedforward architecture, whereas the head direction system is multiple synapses removed and recurrent. The authors find that constructing a working attractor model requires inferring parameters at the level of cell types, with several consistency losses necessary to impose either known biological properties (activity sparsity) or necessary functional properties (correct integration behavior). The authors then demonstrate the robustness of their learned network solution, proposing hypotheses to be tested in future experiments.

Reviewers' responses were mixed. While many found the work well-written and neuroscientifically interesting, there was debate as to the novelty of the approach relative to other connectome-based models, and some reviewers felt that the modeling strategy used to achieve success here may not generalize well to other settings.

During rebuttal, the authors emphasized that this work differs from other connectome-constrained models in its focus on a system quite removed from the sensory periphery and recurrent circuits. Authors conceded that their approach relies heavily on the known function of the circuit, though they also argued that several of their loss objectives could be part of a more general strategy in other circuits, particularly when those circuits involved integrator or attractor behavior. Authors also maintained that while the head direction system is well-studied, their model does offer insights about the system achieves stability and can generate novel experimental predictions. They also noted that this work is distinct from several other noted works in neuroscience using self-supervised learning, which attempt to infer structure (connectivity or cell types) from functional data.

In the end, all reviewers agreed that the work is well-executed and clearly written, and while not all agree about its novelty, the innovative combination of inferring cell-type-specific parameters with self-supervised consistency losses to produce a _functional_ connectome-constrained model is a significant advance. Moreover, while the approach here may be somewhat tailored to the head direction circuit, it contains a number of insights and ingredients that are likely to inform and be adapted to other systems. All these factors argue for acceptance.